# NExUME: Adaptive Training and Inference for DNNs under Intermittent Power Environments

**Cyan Subhra Mishra, Deeksha Chaudhary, Jack Sampson, Mahmut Taylan Knademir, Chita Das**
CSE Department, The Pennsylvania State University

{cyan, dmc6955, jms1257, mtk2, cxd12}@psu.edu

## Abstract

The deployment of Deep Neural Networks (DNNs) in energy-constrained environments, such as Energy Harvesting Wireless Sensor Networks (EH-WSNs), introduces significant challenges due to the intermittent nature of power availability. This study introduces *NExUME*, a novel training methodology designed specifically for DNNs operating under such constraints. We propose a dynamic adjustment of training parameters—dropout rates and quantization levels—that adapt in real-time to the available energy, which varies in energy harvesting scenarios.

This approach utilizes a model that integrates the characteristics of the network architecture and the specific energy harvesting profile. It dynamically adjusts training strategies, such as the intensity and timing of dropout and quantization, based on predictions of energy availability. This method not only conserves energy but also enhances the network's adaptability, ensuring robust learning and inference capabilities even under stringent power constraints. Our results show a 6% to 22% improvement in accuracy over current methods, with an increase of less than 5% in computational overhead. This paper details the development of the adaptive training framework, describes the integration of energy profiles with dropout and quantization adjustments, and presents a comprehensive evaluation using real-world data. Additionally, we introduce a novel dataset aimed at furthering the application of energy harvesting in computational settings.

## 1 Introduction

The increasing demand for ubiquitous, sustainable, and energy-efficient computing, combined with advancements in energy harvesting systems, has spurred significant research into battery-less devices (Gobieski et al., 2019; Resch et al., 2020; Mishra et al., 2021; Saffari et al., 2021; Afzal et al., 2022). Such platforms represent the future of the Internet of Things (IoT) and energy harvesting wireless sensor networks (EH-WSNs). Equipped with modern machine learning (ML) techniques, these devices can revolutionize computing, monitoring, and analytics in remote, risky, and critical environments such as oil wells, mines, deep forests, oceans, remote industries, and smart cities. However, the intermittent and limited energy income of these deployments demands optimizations for ML applications at the algorithm (Yang et al., 2017; Shen et al., 2022; Mendis et al., 2021), orchestration (Maeng & Lucia, 2018; Mishra et al., 2021), compilation (Gobieski et al., 2018), and hardware development (Qiu et al., 2020; Islam et al., 2022; Mishra et al., 2024) layers. Despite these advancements, achieving consistent and accurate inference—thereby meeting service level objectives (SLOs)—in such intermittent environments remains a significant challenge, exacerbated by unpredictable resources, form-factor limitations, and variable computational availability, particularly when employing task-optimized deep neural networks (DNNs).

There are two major problems with performing DNN inference under intermittent power. **(I) Energy Variability**: Even though DNNs can be tailored to match the average energy income of the energy harvesting (EH) source through pruning, quantization, distillation, or network architecture search (NAS) (Yang et al., 2018; 2017; Mendis et al., 2021), there is no guarantee that the energy income consistently meets or exceeds this average. When the income falls below the threshold, the system halts the inference and checkpoints the intermediate states (via software or persistent hardware) (Maeng & Lucia, 2018; Qiu et al., 2020), resuming upon energy recovery. Depending on the EH profile,

this might lead to significant delays and SLO violations. **(II) Computational Approximation**: To address (I) and maintain continuous operation, EH-WSNs may skip some compute during energy shortfalls by dropping neurons (zero padding) or by approximating computations (quantization). Adding further approximation to save energy atop an already heavily reduced network can propagate errors through the layers, leading to significant accuracy drops (Islam & Nirjon, 2019; Kang et al., 2022; Lv & Xu, 2022; Kang et al., 2020), further violating SLOs.

In certain energy-critical scenarios, even EH-WSNs applying state-of-the-art techniques fail to consistently meet SLOs, sometimes skipping entire inferences to deliver results on time. Fundamentally, while current DNNs can be trained or fine-tuned to fit within a given resource budget—be it compute, memory, or energy—they are *not* trained to expect a variable or intermittent resource income. Although intermittency-aware NAS (Mendis et al., 2021), could alleviate certain problems, they often assume fixed resource constraints and do not account for real-time energy fluctuations. Moreover, existing works like Keep in Balance (Yen et al., 2023), Stateful Neural Networks (Yen et al., 2022), ePerceptive (Montanari et al., 2020), and Zygarde (Islam & Nirjon, 2019) address aspects of intermittent computing but do not integrate energy variability awareness directly into the training and inference processes to enable dynamic adaptation. This calls for revisiting the entire training process; we need to train the DNN in such a way that it is aware of the intermittency and *adapts* to it.

Motivated by these challenges, we propose **NExUME** (**N**eural **Ex**ecution **U**nder Inter**M**ittent **E**nvironments), a novel framework designed specifically for environments with intermittent power and EH-WSNs, with potential applications in any ultra-low-power inference system. NExUME uniquely integrates energy variability awareness directly into both the training (**DynFit**) and inference (**DynInfer**) processes, enabling DNNs to dynamically adapt computations based on real-time energy availability. This involves an innovative strategy of learning instantaneous energy-aware dynamic dropout and quantization selection during training, and an intermittency-aware task scheduler during inference. The method includes targeted fine-tuning that not only regularizes the model but also prevents overfitting, enhancing robustness to fluctuations in resource availability. **Our key contributions** can be summarized as follows:

- **DynFit**: A novel training optimizer that embeds energy variability awareness directly into the DNN training process. This optimizer allows for dynamic adjustments of dropout rates and quantization levels based on real-time energy availability, thus maintaining learning stability and improving model accuracy under power constraints.
- **DynInfer**: An intermittency- and platform-aware task scheduler that optimizes computational tasks for intermittent power supply, ensuring consistent and reliable DNN operation. DynInfer leverages software-compiler-hardware co-design to manage and deploy tasks. With the help of DynFit, DynInfer provides 6%–22% accuracy improvements with $\leq 5\%$ additional compute over existing methods.
- **Dataset**: A first-of-its-kind machine status monitoring dataset, involving multiple types of EH sensors mounted at various locations on a Bridgeport machine to monitor its activity status, facilitating research in predictive maintenance and Industry 4.0 applications.

## 2 BACKGROUND AND RELATED WORK

**Energy Harvesting and Intermittent Computing:** The exploding usage of IoTs, connected devices, and wearable electronics project the number of battery operated devices to be 24.1 Billion by 2030 (Insights, 2023). This has a significant economic (users, products and data generating dollar value) as well as environmental (battery and e-waste) impact (Mishra et al., 2024). In fact, advances in EH has lead to a staggering development in intermittently powered battery-free devices (Maeng & Lucia, 2018; Gobieski et al., 2019; Qiu et al., 2020; Saffari et al., 2021; Afzal et al., 2022). A typical EH setup consists of 5 components, namely, energy capture (solar panel, thermocouple, etc), power conditioning, voltage regulation (buck or boost converter), energy storage (super capacitor) and compute unit (refer §Appendix B for details about each of them). To cater towards the sporadic power income and failures, an existing body of works explores algorithms, orchestration, compiler support, and hardware development (Yang et al., 2017; 2018; Mendis et al., 2021; Maeng & Lucia, 2018; Gobieski et al., 2018; Qiu et al., 2020; Islam et al., 2022; Mishra et al., 2024; 2021; Ma et al., 2016; 2017; Liu et al., 2015). Most of these works rely on software checkpointing (static

and dynamic (Maeng & Lucia, 2018), refer §Appendix C) to save and restore, while some of the prior works developed nonvolatile hardware (Ma et al., 2016; 2017) which inherently takes care of the checkpointing. Considering the scope of these initiatives, it is crucial to acknowledge that, despite the substantial support for energy harvesting and intermittency management, developing intermittency-aware applications and hardware necessitates multi-dimensional efforts that span from theoretical foundations to circuit design.

**Intermittent DNN Execution/Training:** As the applications deployed on such EH devices demand analytics, executing DNNs on EH devices and EH-WSNs have become prominent (Lv & Xu, 2022; Gobieski et al., 2019; Qiu et al., 2020; Mishra et al., 2021). However, due to computational constraints, limited memory capacity and restricted operating frequencies, many of these applications fail to complete inference execution with satisfactory SLOs, despite comprehensive software and hardware support (Mishra et al., 2021). While the works relying on loop-decomposition or task partition (e.g., see (Qiu et al., 2020; Gobieski et al., 2019) and the references therein) ensure "forward progress", they do not guarantee an inference completion while meeting SLOs. Optimizing DNNs for the energy constraints (Yang et al., 2018; 2017), or performing early exit and depth-first slicing (Lv & Xu, 2022; Islam & Nirjon, 2019) does ensure more forward progress, but such approaches compromise accuracy while often imposing scheduling overheads and higher memory footprint. One major issue is, most of the works leverage "pre-existing" DNNs, which are typically designed for running on a stable resource environment, while being deployed on an intermittent environment with pseudo notion of stability via check-pointing, and therefore, one direction of works (Mendis et al., 2021) looks for performing network architecture search for intermittent devices. However, this research direction only accounts for fixed lower and upper bounds of energy and compute capacities, overlooking the "sporadic" nature of energy availability and the elasticity of the compute hardware (i.e., the ability to dynamically scale frequency, compute, and memory). Moreover, while the DNN is designed to operate within a specific power window, it is *not* trained to adapt to these fluctuations. Consequently, during extended periods of energy scarcity, the system lacks mechanisms for computational approximation, such as dynamic dropouts (neuron skipping) and dynamic quantization. *Essentially, the DNN is trained to manage within a static resource budget, ignoring the "dynamism" of the resources.* In contrast, our work prioritizes the integration of this dynamism in both the network architecture search (NAS) and the training phases, adapting more effectively to fluctuating energy and compute conditions.

## 3 NExUME FRAMEWORK

To address the issues with *intermittency-aware* DNN training and inference, we propose NExUME: (**N**eural **Ex**ecution **U**nder Inter**M**ittent **E**nvironments). NExUME has three interrelated components: (1) **DynNAS**: Intermittency- and platform-aware neural architecture search; (2) **DynFit**: Intermittency- and platform-aware DNN training with dynamic dropouts and quantization; and (3) **DynInfer**: Intermittency- and platform-aware task scheduling for inference. While each component can individually optimize DNNs for intermittent environments, their combination yields the best results. Our innovation lies in the integration of energy variability awareness directly into both the training and inference processes, enabling dynamic adaptation to real-time energy conditions, which is not addressed by existing methods (Mendis et al., 2021; Yen et al., 2023; 2022; Montanari et al., 2020; Islam & Nirjon, 2019). To search for the best architecture for the given intermittent environment, DynNAS utilizes the approach proposed by iNAS (Mendis et al., 2021). After the network architecture is determined, DynFit is used to train the network considering energy intermittency, and DynInfer is employed to perform inference under intermittent power conditions.

In this section, we elaborate on the key components, focusing on DynFit and DynInfer, and explain how they uniquely adapt DNN training and inference to intermittent power conditions.

### 3.1 DYNFIT: INTERMITTENCY-AWARE LEARNING

**DynFit** is designed to optimize deep neural networks (DNNs) for execution in environments characterized by intermittent power supply due to energy harvesting. The primary goal of DynFit is to adapt the DNN's training process to operate efficiently under unpredictable energy budgets while maintaining acceptable accuracy and adhering to predefined service level objectives (SLOs).

DynFit introduces key mechanisms to dynamically adjust computational complexity based on energy availability, thereby enabling energy-efficient execution of DNN models in constrained environments. These mechanisms include: (i) **Dynamic Dropout**, which adjusts the dropout rates based on available energy to reduce computational load; (ii) **Dynamic Quantization**, which modifies quantization levels in response to energy constraints to save energy; and (iii) **QuantaTask** design, which defines atomic computational units that can be executed without interruption given the energy budget.

Unlike standard implementations where dropout rates and quantization levels are fixed or adjusted solely based on training dynamics, DynFit adjusts these parameters in real-time based on the energy profile of the device. Specifically, during training, we simulate energy variability by incorporating energy traces into the training loop. At each training iteration, the available energy $E_b$ is sampled from these traces. Based on $E_b$, we adjust the dropout rate $d_i$ for each layer $i$ according to:

$$d_i = d_{\max} \left( 1 - \frac{E_b}{E_{\max}} \right), \tag{1}$$

where $d_{\max}$ is the maximum allowable dropout rate, and $E_{\max}$ is the maximum energy observed in the traces. Similarly, the quantization levels $q_j$ are adjusted:

$$q_j = q_{\min} + (q_{\max} - q_{\min}) \frac{E_b}{E_{\max}}. \tag{2}$$

This ensures that when energy is low, higher dropout rates and lower quantization bit-widths are used to reduce computational load, and vice versa.

**Modeling Energy Consumption:** The energy consumption of DNN operations is modeled based on empirical profiling data from the hardware platform. Let $e_{\mathrm{op}}$ denote the energy consumed per computational operation, which varies with operation type and data precision. The total energy consumption of a QuantaTask $q$ is modeled as $E_q = e_{\mathrm{op}} \times \ell_q$, where $\ell_q$ is the number of operations in the task. By integrating the energy model into the training process, DynFit ensures the adjustments to dropout and quantization directly correspond to actual energy savings on the target hardware.

A *QuantaTask* is defined as the smallest atomic unit of computation that can be executed entirely without interruption under the current energy and hardware constraints. Each QuantaTask ensures that execution proceeds without partial computation, which would otherwise lead to overhead from checkpointing and potential data corruption. The main properties of QuantaTasks are atomicity and respect for energy constraints. Figure 1 illustrates QuantaTask execution with a simple example.

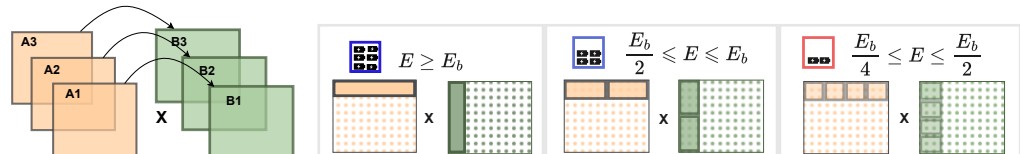

Figure 1: An example of variable QuantaTask in a matrix multiplication scenario. Depending on the available energy, the task (vector inner product) can be divided into multiple iterations such that each QuantaTask is guaranteed to finish given the energy availability. $E$ is available energy, and $E_b$ is the energy required to finish one inner product.

**Optimization Variables, Constraints, and Objective Function:** The optimization problem is formulated with variables: the weights $\mathbf{W}$, dropout rates $\mathbf{d}$, quantization levels $\mathbf{q}$, and QuantaTask sizes $\boldsymbol{\ell}$. The objective is to minimize the total loss, including prediction loss and regularization terms penalizing energy consumption (subject to energy constraints):

$$\min_{\mathbf{W}, \mathbf{d}, \mathbf{q}, \boldsymbol{\ell}} \quad \mathcal{L}(\hat{\mathbf{Y}}, \mathbf{Y}) + \lambda_1 \sum_{j=1}^{M} c_q(q_j) + \lambda_2 \sum_{i=1}^{N} c_d(d_i). \tag{3}$$

**Formulation of the Composite Optimization Problem:** The problem is non-convex due to the discrete nature of quantization levels and dropout rates. We employ an alternating optimization strategy, iteratively optimizing subsets of variables while keeping others fixed. Our method differs from standard approaches by integrating energy constraints directly into the optimization, ensuring that the network learns to adapt its parameters based on energy availability.

### 3.1.1 ADAPTIVE REGULARIZATION STRATEGY

DynFit introduces an adaptive regularization strategy to address potential overfitting and under-training due to uneven weight updates caused by dynamic dropout and quantization. We monitor the update frequency $F_p$ of each weight $w_p$ over a window of $T$ iterations:

$$F_p = \frac{1}{T} \sum_{t=1}^{T} U_p(t), \quad U_p(t) = \begin{cases} 1, & \text{if } w_p \text{ is updated at iteration } t \\ 0, & \text{otherwise} \end{cases} \tag{4}$$

Weights with $F_p < \theta_{\text{low}}$ are considered under-trained, and those with $F_p > \theta_{\text{high}}$ are considered overfitting. We adjust dropout rates and apply L2 regularization accordingly to balance the training process. This adaptive strategy ensures that all weights are adequately trained despite the dynamic adjustments. Dropout scheduling techniques are incorporated, where dropout rates are increased or decreased over time based on the training progress and energy availability, mitigating potential overfitting introduced by static dropout variations.

**Complexity Analysis of DynFit:** The time complexity of DynFit during training is $O(N \cdot T)$, where $N$ is the number of weights and $T$ is the number of training iterations. The overhead introduced by monitoring update frequencies and adjusting dropout rates is negligible compared to the overall training time, as these operations are simple arithmetic computations per iteration. The space complexity is $O(N)$ for storing the update frequencies and additional parameters. Compared to classical training, DynFit adds minimal overhead, with a tradeoff of $\leq 5\%$ additional compute for significant gains in accuracy under intermittent power conditions.

### 3.2 DYNINFER: INTERMITTENCY-AWARE INFERENCE SCHEDULING

**DynInfer** optimizes the inference phase of DNNs operating under intermittent power conditions. Unlike traditional systems with stable power, intermittent environments pose unique challenges for executing inference tasks efficiently and reliably.

The inference process is represented as a set of tasks $\mathcal{T} = \{T_1, T_2, \ldots, T_N\}$, where each task $T_i$ is characterized by its energy requirement $E_i$, execution time $\tau_i$, priority $p_i$, deadline $D_i$, and criticality level $c_i$. At any given time $t$, the available energy is denoted as $E_b(t)$.

**Task Fusion and Scheduling:** DynInfer introduces a novel task scheduling algorithm that dynamically adjusts to real-time energy availability. When the energy required for executing multiple QuantaTasks exceeds the available energy budget, DynInfer employs *task fusion* to combine smaller tasks into larger atomic units that can be executed within the energy constraints.

**Formal Definition of Task Fusion:** Let $\mathcal{Q} = \{q_1, q_2, \ldots, q_k\}$ be a set of QuantaTasks with individual energy requirements $E_{q_i}$. If $\sum_i E_{q_i} \leq E_b$, the available energy budget, then tasks can be executed sequentially without interruption. However, if $\sum_i E_{q_i} > E_b$, we aim to fuse tasks to minimize checkpointing overhead. Task fusion is formalized as finding a partition of $\mathcal{Q}$ into subsets $\mathcal{Q}_1, \mathcal{Q}_2, \ldots, \mathcal{Q}_m$ such that, for each subset $\mathcal{Q}_j$, $\sum_{q_i \in \mathcal{Q}_j} E_{q_i} \leq E_b$, and $m$ is minimized. This reduces the number of checkpoints and the overhead associated with task switching. For example, Consider two convolution operations $C_1$ and $C_2$ with energy requirements $E_{C_1}$ and $E_{C_2}$, respectively. If individually $E_{C_1}, E_{C_2} > E_b$ but $E_{C_1} + E_{C_2} \leq E_b$, we fuse $C_1$ and $C_2$ into a single task. The fused task executes both convolutions atomically within the energy budget, avoiding the overhead of checkpointing between them.

**Scheduling Problem Formulation:** The scheduling problem is formulated with decision variables $s_i$ (task start times) and binary variables $x_i \in \{0, 1\}$ (indicating whether a task is scheduled). The energy availability constraint over time is expressed as (subject to energy and task constraints): $\sum_{i:s_i \leq t < f_i} E_i \leq E_b(t)$ The objective is to maximize the total weighted priority of scheduled tasks:

$$\max_{\{x_i, s_i\}} \quad \sum_{i=1}^{N} \left( p_i - \alpha E_i - \beta (f_i - D_i)^+ \right) x_i. \tag{5}$$

**Scheduling Performance Assurance:** Our scheduling heuristic, *Energy-Aware Priority Scheduling*, while sub-optimal in the theoretical sense, is designed to perform near-optimally in practice for real-time systems. We ensure its performance by: 1. *Empirical Validation*: We compare the heuristic's

performance with the optimal solution on smaller problem instances using exhaustive search and find that the heuristic achieves within 95% of the optimal task completion rate. 2. *Theoretical Analysis*: The heuristic prioritizes tasks based on effective priority $P_i^{\text{eff}} = \frac{p_i}{E_i} \times \phi_i$, where $\phi_i$ accounts for deadline urgency. This balances task importance against energy consumption, leading to efficient utilization of available energy. 3. *Complexity Analysis*: The heuristic has a time complexity of $O(N \log N)$ due to sorting tasks based on $P_i^{\text{eff}}$, which is acceptable for real-time applications.

**Complexity Analysis of DynInfer:** The time complexity of the scheduling algorithm is $O(N \log N)$ due to sorting tasks, and the space complexity is $O(N)$ for storing task parameters. Compared to classical inference, DynInfer introduces additional overhead for scheduling and task fusion, but this is offset by the gains in reliability and efficiency under intermittent power.

**Handling Extremely Low or Sporadic Energy Levels:** In environments with extremely low or sporadic energy levels where consistent dropout and quantization adjustments may not be feasible, NExUME handles this by: 1. Implementing a minimum viable model configuration that operates at the lowest acceptable energy consumption, achieved by maximizing dropout rates and using the lowest quantization bit-widths. 2. Prioritizing essential tasks and deferring non-critical computations. 3. Employing predictive energy harvesting models to anticipate energy availability and adjust computations proactively. In extreme cases, the system can enter into a low-power standby mode and resume operation when sufficient energy is available. These strategies ensure that the system remains operational and provides degraded but acceptable performance under severe energy constraints.

**Novelty in Energy-Aware Scheduling:** While energy-aware scheduling is not novel in itself, our contribution lies in adapting scheduling algorithms specifically for intermittent power environments. Existing scheduling algorithms typically assume stable energy availability and do not account for the atomicity constraints imposed by intermittent power supply. Our scheduling approach uniquely integrates: 1. Real-time energy availability into scheduling decisions. 2. Task fusion to minimize checkpointing overhead, which is critical in intermittent environments. 3. Dynamic adjustment of computational tasks based on both energy and task criticality. These innovations enable efficient and reliable DNN inference under intermittent power conditions, differentiating our work from existing energy-aware schedulers.

**Rationale Behind Method Design:** The overall method design of NExUME is motivated by the need to enable DNNs to function reliably in environments with intermittent and unpredictable energy supply. By integrating energy variability into both training and inference, we allow the DNN to adapt its computational load dynamically, ensuring that critical tasks are completed within energy constraints. This holistic approach addresses the limitations of existing methods that treat training and inference separately or do not account for real-time energy fluctuations.

**Implementation Details:** We propose a software-compiler-hardware co-designed framework for devices with non-volatility (e.g., MSP-EXP430FR5994 with FeRAM). Figure 2 outlines our design. User programs (P1) are supported by a moving-window power predictor (P2) that uses the EH capacitor input to decide execution based on available energy. The compiler decomposes the program into a DAG of jobs (e.g., CONV2D (C1), batch normalization (C2)). Larger tasks are profiled on the MSP-EXP430FR5994, split into Power Atomic Tasks (QuantaTasks), and optimized in assembly. NV FeRAM is used for backup and restore during power emergencies.

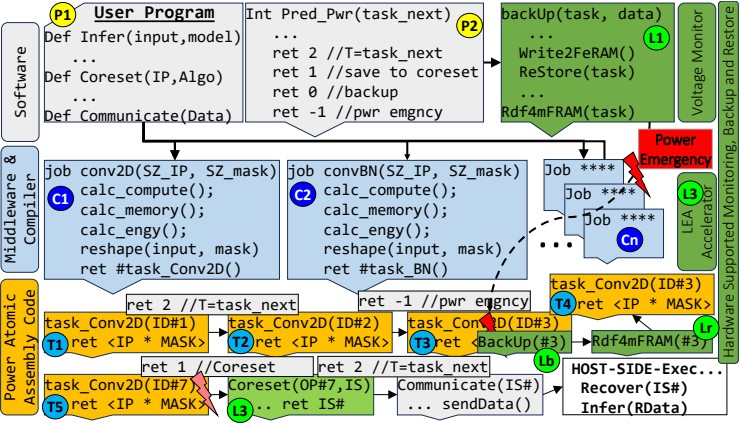

Figure 2: Software-Compiler-Hardware Driven DynInfer Flow.

# 4 EXPERIMENTAL RESULTS

NExUME can be seamlessly integrated as a "plug-in" for *both* training and inference frameworks in deep neural network (DNN) applications, specifically designed for intermittent and (ultra) low-power deployments. In this section, we discuss the effectiveness of NExUME across two distinct types of environments, highlighting its versatility and broad applicability. Firstly, we evaluate NExUME using publicly available datasets (§4.2) commonly utilized in embedded applications across multiple modalities—including image, time series sensor, and audio data. These datasets represent typical use cases in embedded systems where energy efficiency and minimal computational overhead are crucial. We use both commercial-off-the-shelf (COTS) hardware and state-of-the-art ReRAM Xbar-based hardware for this evaluation. Secondly, we introduce a novel dataset aimed at advancing research in predictive maintenance and Industry 4.0 (Lasi et al., 2014), and test NExUME on a real manufacturing testbed (§4.3) with COTS hardware. We have developed a first-of-its-kind machine status monitoring dataset, available at `https://hackmd.io/@Galben/rk7YN6jmR`, which involves mounting multiple types of sensors at various locations on a Bridgeport machine to monitor its activity status.

## 4.1 DEVELOPMENT AND PROFILING OF NExUME

NExUME uses a combination of programming languages and technologies to optimize its functionality in intermittent and low-power computing environments. The software stack comprises Python3 (2.7k lines of code), CUDA (1.1k lines of code), and Embedded C (2.1k lines of code, not including DSP libraries). Our training infrastructure utilizes NVIDIA A6000 GPUs with 48 GiB of memory, supported by a 24-core Intel Xeon Gold 6336Y CPU. We employ PyTorch v2.3.0 coupled with CUDA version 11.8 as our primary training framework. To assess the computational overhead introduced by DynFit, a component of NExUME, we use NVIDIA Nsight Compute. During the training sessions enhanced by DynFit, we observed an increase in the number of instructions ranging from a minimum of 11.4% to a maximum of 34.2%. While the overhead in streaming multi-processor (SM) utilization was marginal (within 5%), there was a noticeable increase in memory bandwidth usage, ranging from 6% to 17%. Moreover, we have implemented a modified version of the matrix multiplication operation that strategically skips the loading of rows and/or columns from the input matrices into the GPU's shared memory and register files. This adaptation is guided by the dropout mask vector and the specific type of sparse matrix operation being performed. This technique effectively reduces the number of load operations by an average of 12%, thereby enhancing the efficiency of computations under energy constraints and contributing to the overall performance improvements in NExUME.

## 4.2 NExUME ON PUBLICLY AVAILABLE DATASETS

**Datasets:** For image data, we consider the Fashion-MNIST (Xiao et al., 2017) and CIFAR10 (Alex, 2009) datasets; for time series sensor data, we focus on popular human activity recognition (HAR) datasets, MHEALTH (Banos et al., 2014) and PAMAP2 (Reiss & Stricker, 2012); and for audio, we use the AudioMNIST (Becker et al., 2023) dataset.

**Inference Deployment Embedded Platforms:** For commercially off-the-shelf micro-controllers, we choose Texas Instruments MSP430FR5994 (Instruments, 2024a), and Arduino Nano 33 BLE Sense (Arduino, 2024) as our deployment platforms with a Pixel-5 phone as the host device. The host device is used for data logging—collecting SLOs, violations, power failures, etc., along with running the "baseline" inferences without intermittency.

**Baselines:** We take the combination of best available approaches for DNN inference on intermittent environment as baselines. All these DNNs are executed with the state-of-the-art checkpointing and scheduling approach (Maeng & Lucia, 2018). Baseline **Full Power** is a DNN designed by iNAS (Mendis et al., 2021) for running while the system is battery-powered and has to hit a target SLO (latency < 500ms). Baseline **AP** is a DNN compressed to fit the average power of the energy harvesting (EH) environment using iNAS (Mendis et al., 2021) and energy-aware pruning (EAP) (Yang et al., 2017; 2018). Baseline **PT** takes the **Full Power** DNN and uses techniques proposed by (Yang et al., 2018) and (Yang et al., 2017) to prune, quantize, and compress the model. Baseline **iNAS+PT** designs the network from the ground up while combining the work of iNAS (Mendis et al., 2021) and EAP (Yang et al., 2018; 2017).

We also compare our approach with recent state-of-the-art methods specifically designed for intermittent systems, namely **Stateful** (Yen et al., 2022), **ePerceptive** (Montanari et al., 2020), and **DynBal** (Yen et al., 2023). These methods introduce various techniques such as embedding state information into the DNN, multi-resolution inference, multi-exit architectures, and runtime reconfigurability to handle intermittency in energy-harvesting devices. We have faithfully re-implemented these methods as per the descriptions and adjusted them for a fair comparison under our setup.

**Results:** Table 1 shows the accuracy of our approach against the baselines and the recent state-of-the-art methods using the TI MSP board powered by piezoelectric energy harvesting. The inferences meeting the SLO requirements are the only ones considered for accuracy; i.e., a correct classification violating the latency SLO is considered as "incorrect".

| Datasets | Full Power | AP | PT | iNAS+PT | Stateful | ePerceptive | DynBal | NExUME |
|---|---|---|---|---|---|---|---|---|
| FMNIST | 98.70 | 71.90 | 79.72 | 83.68 | 85.40 | 86.25 | 87.50 | **88.90** |
| CIFAR10 | 89.81 | 55.05 | 62.00 | 66.98 | 68.50 | 70.20 | 71.75 | **76.29** |
| MHEALTH | 89.62 | 59.76 | 65.40 | 71.56 | 73.80 | 74.95 | 76.10 | **80.75** |
| PAMAP | 87.30 | 57.38 | 65.77 | 70.33 | 72.20 | 73.35 | 74.50 | **75.16** |
| AudioMNIST | 88.20 | 67.29 | 73.16 | 75.41 | 76.80 | 77.95 | 78.60 | **80.01** |

Table 1: Accuracy comparison on TI MSP board using piezoelectric energy harvesting.

As observed in Table 1, NExUME consistently outperforms the state-of-the-art methods across all datasets. For instance, on CIFAR10, NExUME achieves an accuracy of 76.29%, which is approximately 4.54% higher than DynBal, the next best method. This improvement is significant in the context of energy-harvesting intermittent systems, where achieving high accuracy under strict energy constraints is challenging. The superior performance of NExUME can be attributed to its unique integration of energy variability awareness directly into both the training (DynFit) and inference (DynInfer) processes. Unlike other methods that either focus on modifying the DNN architecture or optimizing inference configurations, NExUME adapts the DNN's computational complexity in real-time based on instantaneous energy availability, leading to more efficient use of scarce energy resources and improved accuracy.

| Dataset | Platform | Energy Source | Stateful | ePerceptive | DynBal | NExUME |
|---|---|---|---|---|---|---|
| FMNIST | MSP430FR5994 | Piezoelectric | 20.1 | 20.8 | 21.5 | **23.4** |
| CIFAR10 | Arduino Nano | Thermal | 16.0 | 16.5 | 17.0 | **18.5** |
| MHEALTH | ESP32 S3 Eye | Piezoelectric | 18.5 | 19.0 | 19.6 | **21.0** |
| PAMAP | STM32H7 | Thermal | 16.5 | 17.0 | 17.5 | **19.0** |
| AudioMNIST | Raspberry Pi Pico | Piezoelectric | 20.5 | 21.0 | 21.7 | **23.2** |

Table 2: Energy efficiency comparison on different hardware platforms.

Table 2 presents the energy efficiency in MOps/Joule for each dataset on different hardware platforms using piezoelectric and thermal energy harvesting. NExUME achieves the highest energy efficiency across all platforms and datasets. This demonstrates that NExUME not only improves accuracy but also enhances energy utilization, making it highly suitable for deployment in energy-constrained intermittent environments. The improvements in energy efficiency are due to NExUME's ability to adjust computational workload dynamically, minimizing energy wastage and ensuring that computations are matched to the available energy budget. NExUME, thanks to its inherent learnt adaptability, significantly reduces saves, restores, reconfigurations and READ/WRITE from/to nonvolatile memory or to the flash memory in the cases and devices where NVMs are not present which gives it edge over the baselines across multiple devices.

**Discussion of Results:** 1. *Dynamic Adaptation:* NExUME's DynFit and DynInfer components enable real-time adjustments of dropout rates and quantization levels during training and inference based on instantaneous energy availability. This allows the DNN to maintain high accuracy even under severe energy constraints. 2. *Energy Variability Awareness:* By integrating energy profiles directly into the training process, NExUME ensures that the model learns to handle fluctuations in energy supply, leading to more robust performance compared to methods that do not consider energy variability during training. 3. *Efficient Scheduling:* DynInfer's energy-aware task scheduling and task

fusion mechanisms reduce overhead from checkpointing and optimize the execution of tasks within the available energy budget. 4. *Holistic Approach:* Unlike other methods that focus on either training or inference optimizations, NExUME provides a comprehensive solution that addresses both phases, leading to superior overall performance.

### 4.3 NEXUME ON MACHINE STATUS MONITORING *[Our New Dataset]*

Automation and monitoring and analytics are the key ingredients in the upcoming Industry 4.0. To enable sustainable machine status monitoring with energy harvesting (from machine vibrations or Wifi signals) we evaluate our setup using Bridgeport machines for monitoring their status. Prior works (Center, 2018) majorly focused on fault analysis but there are little to no datasets on predictive maintenance. **Setup and Sensor Arrangement:** Two different types of 3-axis accelerometers (with 100Hz and 200Hz sampling rate) were placed in three different locations of a Bridgeport machine to collect and analyze data under different operating status. There were 5 operating statuses: three different speeds of rotation of the spindle (**R1: 100RPM**, **R2: 200RPM**, **R3: 300RMP** with no job; RPM – rotations per minute), spindle under job (**SJ**), and spindle idle (**SI**). We collected over 700,000 samples over a period of 2 hours for each of the sensors. The sensor data were cleaned, normalized, and converted to the power spectrum density for further analysis. We use iNAS (Mendis et al., 2021) to find the DNNs meeting the energy income and train them using our proposed DynFit. Table 3 shows the accuracy of classification tasks against the different baselines and state-of-the-art methods.

| Class | Full Power | AP | PT | iNAS+PT | Stateful | ePerceptive | DynBal | NExUME |
|-------|-----------|-------|-------|---------|----------|-------------|--------|--------|
| **R1** | 84.93 | 74.46 | 77.02 | 79.62 | 80.85 | 81.50 | 82.15 | **83.60** |
| **R2** | 85.85 | 76.21 | 79.18 | 80.36 | 81.95 | 82.60 | 83.25 | **84.50** |
| **R3** | 81.09 | 72.43 | 75.38 | 78.18 | 79.05 | 79.70 | 80.35 | **80.85** |
| **SJ** | 90.95 | 82.33 | 85.00 | 87.58 | 88.60 | 89.15 | 89.80 | **90.50** |
| **SI** | 94.76 | 85.31 | 88.05 | 89.90 | 91.00 | 91.65 | 92.30 | **93.00** |

Table 3: Accuracy of NExUME and other methods for industry status monitoring dataset using TI MSP board and piezoelectric energy source. Results collected over 200 experiment cycles.

NExUME demonstrates superior performance across all operating classes, achieving the highest accuracy in each case. For example, for the spindle idle (SI) class, NExUME attains an accuracy of 93.00%, outperforming DynBal by 0.70%. While the margins may appear small, in industrial settings, even minor improvements in classification accuracy can have significant implications for predictive maintenance and operational efficiency. The improved performance of NExUME in this real-world application further validates its effectiveness and practical utility. By effectively managing energy constraints and adapting to intermittent power conditions, NExUME enables more reliable and accurate monitoring in industrial environments where energy harvesting is a viable power solution.

### 4.4 SENSITIVITY AND ABLATION STUDIES OF NEXUME

To elucidate the influence of variable SLOs and hardware-specific settings on system performance, we conducted a comprehensive sensitivity study. This study involved adjusting the acceptable latency and the capacitance of the energy harvesting setup to assess their impacts on accuracy. As shown in Figure 3a, the accuracy improves with increased latency, but with diminishing returns. Similarly, Figure 3b demonstrates that, while increasing capacitance should theoretically stabilize the system, its charging characteristics can lead to extended charging times, thus exceeding the latency SLO. Notably, some anomalies in the data were attributed to abrupt power failures, a common challenge in intermittent energy harvesting systems. An ablation study evaluates the contributions of individual components within NExUME. The results, plotted in Figure 3c, indicate that the greatest improvements are derived from the "synergistic operation" of all components, particularly DynFit and DynInfer. Although iNAS enhances network selection, its lack of intermittency awareness significantly impacts accuracy.

### 4.5 LIMITATIONS AND DISCUSSION

We recognize that modern architectures like Transformers have become prevalent in the ML community due to their superior performance on large-scale datasets. However, deploying such architectures

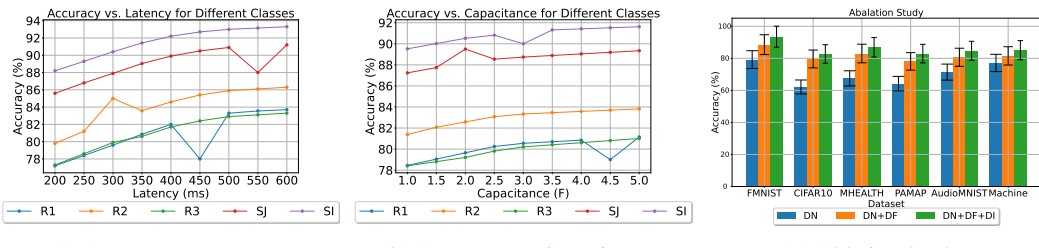

(a) Accuracy vs Latency     (b) Accuracy vs Capacitance     (c) Ablation Study

Figure 3: Sensitivity and ablation study. DN is DynNAS, DF is DynFit, and DI is DynInfer.

on ultra-low-power, energy-harvesting devices presents significant challenges due to their substantial computational and memory requirements. NExUME focuses on enabling efficient and reliable deployment of DNNs in intermittent environments, which are often constrained in terms of computational resources and energy availability. In many real-world applications, especially in IoT and edge computing, there is a critical need for smaller, energy-efficient models that can operate autonomously without reliance on batteries. These tiny, reusable devices contribute to reducing embodied carbon and represent a significant step toward sustainability. Moreover, we believe that advancing the capabilities of smaller models in intermittent environments is crucial for widespread adoption of sustainable, battery-free devices in various domains, including environmental monitoring, industrial IoT, and healthcare. By addressing the challenges of intermittent computing, our work contributes to the broader goal of enabling pervasive, sustainable intelligence at the edge.

NExUME is especially advantageous in intermittent environments, and its utility extends to ultra-low-power or energy scavenging systems. However, the efficacy of DynFit and iNAS is contingent upon the breadth and depth of the available dataset. Additionally, profiling devices to ascertain their energy consumption, computational capabilities, and memory footprint necessitates detailed micro-profiling using embedded programming. This process, while informative, yields only approximate models that are inherently prone to errors. DynFit, with its stochastic dropout features, occasionally leads to overfitting, necessitating meticulous fine-tuning. While effective in smaller networks, our studies involving larger datasets (such as ImageNet) and more complex network architectures (like MobileNetV2 and ResNet) reveal challenges in achieving convergence without precise fine-tuning. DynFit tends to introduce multiple intermediate states during the training process, resulting in approximately 14% additional wall-time on average. The development of DynInfer requires an in-depth understanding of microcontroller programming and compiler directives. The absence of comprehensive library functions along with the need for computational efficiency frequently necessitates the development of in-line assembly code for certain computational kernels.

## 5 CONCLUSIONS

This study presents NExUME, an advanced framework designed to optimize the training and inference phases of deep neural networks within the constraints of intermittently powered, energy-harvesting devices. By integrating adaptive neural architecture and energy-aware training techniques, NExUME significantly enhances the viability of deploying machine learning models in environments with limited and unreliable energy sources. The results from our extensive evaluations demonstrate that NExUME can substantially outperform traditional methods in energy-constrained settings, with improvements in accuracy and efficiency that facilitate real-world applications in remote and wearable technology. Specifically, improvements ranging from 6.10% to 17.13% over existing methods highlight NExUME's capability to adapt dynamically to fluctuating energy conditions, ensuring both operational longevity and computational integrity. The broader implication of this work extends beyond technological advancements, suggesting a paradigm shift in how the machine learning community approaches the design and deployment of systems in energy-limited environments. By prioritizing energy efficiency and system adaptability, NExUME contributes to the sustainability and accessibility of machine learning solutions, enabling their deployment in regions where power infrastructure is absent or unreliable. This is particularly crucial in developing regions where such technology can drive innovation in healthcare, agriculture, and education. Furthermore, the development of energy-efficient, adaptive systems like NExUME is aligned with the growing need for sustainable computing practices across all disciplines of technology. It challenges the machine learning community to consider not only the accuracy and efficiency of algorithms but also their environmental impact and accessibility, ensuring a broader positive social impact.

ACKNOWLEDGMENTS

We sincerely thank the anonymous reviewers for their insightful comments, which have significantly improved this paper. This work was supported in part by Semiconductor Research Corporation (SRC), the Center for Brain-inspired Computing (C-BRIC), NSF Grant #1822923, NSF Grant #2346953, and the Clean Energy Smart Manufacturing Innovation Institute award #136067. We also gratefully acknowledge the Factory for Advanced Manufacturing Education Lab at The Pennsylvania State University for their support in data collection on the Bridgeport machine. All product names are used solely for identification purposes and may be trademarks of their respective companies.

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

APPENDIX

## A MORE RESULTS ON OTHER PLATFORMS AND EH SOURCES

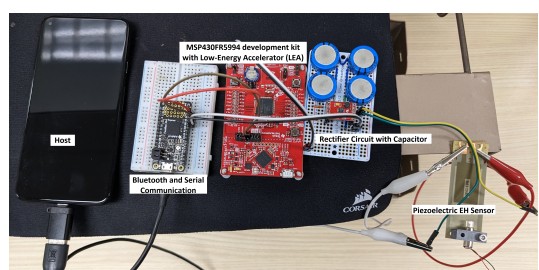

Figure 4: Hardware setup of NExUME using MSP-EXP430FR5994 as the edge compute, Adafruit ItsyBitsy nRF52840 Express for communicating, Energy Harvester Breakout - LTC3588 with super-capacitors as energy rectification and storage and a Pixel-5 phone as the host.

| Datasets | Full Power | MSP on Piezo | | | | |
|---|---|---|---|---|---|---|
| | | AP | PT | iNAS+PT | NExUME | Better |
| **FMNIST** | 98.70 | 71.90 | 79.72 | 83.68 | **88.90** | 6.24% |
| **CIFAR10** | 89.81 | 55.05 | 62.00 | 66.98 | **76.29** | 13.90% |
| **MHEALTH** | 89.62 | 59.76 | 65.40 | 71.56 | **80.75** | 12.84% |
| **PAMAP** | 87.30 | 57.38 | 65.77 | 65.38 | **75.16** | 14.97% |
| **AudioMNIST** | 88.20 | 67.29 | 73.16 | 75.41 | **80.01** | 6.10% |

Table 4: Accuracy of NExUME on MSP board using vibration from a Piezoelectric harvestor. Better refers to the improvement over iNAS+PT baseline.

| Datasets | Full Power | MSP on Thermal | | | | |
|---|---|---|---|---|---|---|
| | | AP | PT | iNAS+PT | NExUME | Better |
| **FMNIST** | 98.70 | 80.92 | 86.32 | 88.93 | **95.62** | 7.53% |
| **CIFAR10** | 89.81 | 64.78 | 69.29 | 71.53 | **83.78** | 17.13% |
| **MHEALTH** | 89.62 | 69.77 | 73.99 | 77.70 | **89.62** | 15.34% |
| **PAMAP** | 87.30 | 66.33 | 71.84 | 74.47 | **85.24** | 14.46% |
| **AudioMNIST** | 88.20 | 73.84 | 78.03 | 81.60 | **87.64** | 7.40% |

Table 5: Accuracy of NExUME on MSP board using thermocouple based thermal harvester. Better refers to the improvement over iNAS+PT baseline.

| Datasets | Full Power | Arduino on RF | | | | |
|---|---|---|---|---|---|---|
| | | AP | PT | iNAS+PT | NExUME | Better |
| **FMNIST** | 98.70 | 74.44 | 79.63 | 83.61 | **90.44** | 8.17% |
| **CIFAR10** | 89.81 | 58.11 | 63.91 | 65.01 | **79.60** | 22.44% |
| **MHEALTH** | 89.62 | 63.52 | 67.40 | 74.30 | **83.86** | 12.87% |
| **PAMAP** | 87.30 | 61.39 | 67.24 | 69.45 | **77.00** | 10.87% |
| **AudioMNIST** | 88.20 | 66.11 | 74.28 | 76.60 | **78.87** | 2.97% |

Table 6: Accuracy of NExUME on Arduino nano board using WiFi based RF harvester. Better refers to the improvement over iNAS+PT baseline.

| Datasets | Full Power | Arduino on Thermal | | | | |
|----------|-----------|------|------|---------|---------|--------|
| | | AP | PT | iNAS+PT | NExUME | Better |
| **FMNIST** | 98.70 | 77.04 | 80.44 | 83.08 | **89.90** | 8.20% |
| **CIFAR10** | 89.81 | 60.38 | 65.90 | 66.98 | **80.70** | 20.48% |
| **MHEALTH** | 89.62 | 65.74 | 69.88 | 72.41 | **85.75** | 18.42% |
| **PAMAP** | 87.30 | 62.76 | 65.93 | 71.46 | **81.27** | 13.73% |
| **AudioMNIST** | 88.20 | 69.12 | 73.86 | 77.79 | **83.54** | 7.39% |

Table 7: Accuracy of NExUME on Arduino nano board using thermocouple based thermal harvester. Better refers to the improvement over iNAS+PT baseline.

## B  DETAILS ON ENERGY HARVESTING

A typical energy harvesting (EH) setup captures and converts environmental energy into usable electrical power, which can then support various electronic devices. Here's a simplified breakdown of the process:

1. **Energy Capture**: The setup begins with a harvester, such as a solar panel, piezoelectric sensor, or thermocouple. These devices are designed to collect energy from their surroundings—light, mechanical vibrations, or heat, respectively.

2. **Power Conditioning**: Once energy is harvested, it often needs to be converted and stabilized for use. This is done using a rectifier, which transforms alternating current (AC) into a more usable direct current (DC).

3. **Voltage Regulation**: After rectification, the power might not be at the right voltage for the device it needs to support. A matching circuit, including components like buck or boost converters, adjusts the voltage to the appropriate level, ensuring the device receives the correct current and voltage.

4. **Energy Storage**: Finally, to ensure a continuous power supply even when the immediate energy source is inconsistent (like when a cloud passes over a solar panel), the system includes a temporary storage unit, such as a super-capacitor. This component helps smooth out the supply, providing steady power to the compute circuit.

By integrating these components, an EH system can sustainably power devices without relying on traditional power grids, making it ideal for remote or mobile applications.

## C  INTERMITTENT COMPUTING AND CHECK-POINTING

### C.1  INTERMITTENCY-AWARE GENERAL MATRIX MULTIPLICATION (GEMM)

Here we explain the operation of an energy-aware algorithm for performing General Matrix Multiplication (GeMM). The algorithm is designed to operate in environments where energy availability is intermittent, such as in devices powered by energy harvesting. It includes mechanisms for loop tiling, checkpointing, and resumption to manage computation across power interruptions effectively.

#### C.1.1  ALGORITHM OVERVIEW

The GeMM operation, typically expressed as $C = A \times B$, where $A$, $B$, and $C$ are matrices, is implemented with considerations for energy limitations. The algorithm breaks the matrix multiplication into smaller chunks (tiles), periodically saves the state before potential power losses, and resumes computation from the last saved state upon power restoration.

#### C.1.2  FUNCTION DEFINITIONS

- **SAVE_STATE**: Saves the current indices and the partial result of the output matrix $C$ to non-volatile memory to allow recovery after a power interruption.

- **LOAD_STATE**: Retrieves the last saved indices and partial result from non-volatile memory to resume computation.

### C.1.3 LOOP TILING

The algorithm uses loop tiling to divide the computation into smaller blocks that can be managed between power interruptions. This tiling not only makes the computation manageable but also optimizes memory usage and cache performance, which is critical in constrained environments.

### C.1.4 CHECK-POINTING MECHANISM

Before each power interruption, detected through an energy monitoring system, the algorithm saves the current state using the **SAVE_STATE** function. This state includes the loop indices and the current value of the element being processed in $C$. This ensures that no computation is lost when the power goes out.

### C.1.5 RESUMPTION MECHANISM

Upon resuming, the algorithm loads the saved state using the **LOAD_STATE** function. This state is used to continue the computation exactly where it left off, minimizing redundant operations and ensuring efficiency.

## D FORMULATION OF DYNAMIC DROPOUTS:

### D.1 L2 DYNAMIC DROPOUT WITH QUANTATASK OPTIMIZATION

L2 Dynamic Dropout leverages the L2 norm of the weights to influence dropout rates, combined with the QuantaTask optimization to handle energy constraints in intermittent systems.

**Mathematical Formulation:** Let $\mathbf{W}$ be the weight matrix of a layer. The L2 norm of the weights is calculated as:

$$\|\mathbf{W}\|_2 = \sqrt{\sum_{i,j} W_{ij}^2}$$

Define the dropout probability $p_i$ for neuron $i$ based on the L2 norm of its corresponding weights. The idea is to use the inverse of the L2 norm to determine the probability:

$$p_i = \frac{\alpha}{\|\mathbf{W}_i\|_2 + \epsilon}$$

where $\alpha$ is a scaling factor to adjust the overall dropout rate, and $\epsilon$ is a small constant to avoid division by zero. Define a binary dropout mask $\mathbf{m} = [m_1, m_2, \ldots, m_n]$ where $m_i \in \{0, 1\}$. Each element of the mask is determined by sampling from a Bernoulli distribution with probability $1 - p_i$:

$$m_i \sim \text{Bernoulli}(1 - p_i)$$

Apply the dropout mask during the forward pass. Let $\mathbf{a}_i$ denote the activation of neuron $i$:

$$\mathbf{a}_i^{\text{dropout}} = \mathbf{a}_i \cdot m_i$$

**Training with L2 Dynamic Dropout and QuantaTask Optimization:** Initialize the network parameters $\mathbf{W}$, dropout mask $\mathbf{m}$, and scaling factor $\alpha$. Define the energy budget $E_b$ for a single quanta and for the entire inference. Initialize the loop iteration parameters $l$. Compute the activations $\mathbf{a}$ and apply the dropout mask:

$$\mathbf{a}_i^{\text{dropout}} = \mathbf{a}_i \cdot m_i$$

Compute the loss $\mathcal{L}(\mathbf{Y}, \hat{\mathbf{Y}})$ where $\mathbf{Y}$ is the output of the network and $\hat{\mathbf{Y}}$ is the target output. Calculate the gradients of the loss with respect to the weights:

$$\frac{\partial \mathcal{L}}{\partial W_{ij}}$$

For each layer $L$ and loop $i$ within the layer, estimate the energy $E_i$ required for the current quanta size $l_i$:

$$E_i \leftarrow \text{DynAgent.estimateEnergy}(L, i, l_i)$$

If $E_i > E_b$, fuse tasks to reduce the overhead:

$$\text{FuseTasks}(L, i, l_i, E_b)$$

Update $E_i$ after task fusion:

$$E_i \leftarrow \text{DynAgent.estimateEnergy}(L, i, l_i)$$

Update the dropout mask $\mathbf{m}$ based on the L2 norm of the weights:

$$p_i = \frac{\alpha}{\|\mathbf{W}_i\|_2 + \epsilon}$$

$$m_i = \begin{cases} 0 & \text{if Bernoulli}(1 - p_i) = 0 \\ 1 & \text{otherwise} \end{cases}$$

Perform the backward pass to update the network weights, considering the dropout mask:

$$\mathbf{W} \leftarrow \mathbf{W} - \eta \frac{\partial \mathcal{L}}{\partial \mathbf{W}} \odot \mathbf{m}$$

where $\eta$ is the learning rate and $\odot$ denotes element-wise multiplication.

**Inference with L2 Dynamic Dropout and QuantaTask Optimization:** Check the available energy using DynAgent. If energy is below a threshold, increase the dropout rate to ensure the inference can be completed within the energy budget. Otherwise, maintain or reduce the dropout rate to improve accuracy. Perform the forward pass with the updated dropout mask to obtain the output $\mathbf{Y}$. This approach ensures that the network is robust to varying energy conditions by incorporating dynamic dropout influenced by the L2 norm of the weights, along with the QuantaTask optimization to handle energy constraints.

### D.2 OPTIMAL BRAIN DAMAGE DROPOUT WITH QUANTATASK OPTIMIZATION

Optimal Brain Damage Dropout leverages a simplified version of the Optimal Brain Damage pruning method to adjust dropout rates, combined with the QuantaTask optimization to handle energy constraints in intermittent systems.

**Mathematical Formulation:** Let $\mathbf{W}$ be the weight matrix of a layer. The sensitivity of each weight $W_{ij}$ is calculated using the second-order Taylor expansion of the loss function $\mathcal{L}$:

$$\Delta \mathcal{L} \approx \frac{1}{2} \sum_{i,j} \frac{\partial^2 \mathcal{L}}{\partial W_{ij}^2} (W_{ij})^2$$

where $\frac{\partial^2 \mathcal{L}}{\partial W_{ij}^2}$ is the second-order derivative (Hessian) of the loss with respect to the weights.

Define the dropout probability $p_i$ for neuron $i$ based on the sensitivity of its corresponding weights. The idea is to use the sensitivity to determine the probability:

$$p_i = \frac{\beta \sum_j \frac{\partial^2 \mathcal{L}}{\partial W_{ij}^2} (W_{ij})^2}{\max \left( \sum_j \frac{\partial^2 \mathcal{L}}{\partial W_{ij}^2} (W_{ij})^2 \right) + \epsilon}$$

where $\beta$ is a scaling factor to adjust the overall dropout rate, and $\epsilon$ is a small constant to avoid division by zero.

Define a binary dropout mask $\mathbf{m} = [m_1, m_2, \ldots, m_n]$ where $m_i \in \{0, 1\}$. Each element of the mask is determined by sampling from a Bernoulli distribution with probability $1 - p_i$:

$$m_i \sim \text{Bernoulli}(1 - p_i)$$

Apply the dropout mask during the forward pass. Let $\mathbf{a}_i$ denote the activation of neuron $i$:

$$\mathbf{a}_i^{\text{dropout}} = \mathbf{a}_i \cdot m_i$$

**Training with Optimal Brain Damage Dropout and QuantaTask Optimization:** Initialize the network parameters $\mathbf{W}$, dropout mask $\mathbf{m}$, and scaling factor $\beta$. Define the energy budget $E_b$ for a single quanta and for the entire inference. Initialize the loop iteration parameters $l$.

Compute the activations $\mathbf{a}$ and apply the dropout mask:

$$\mathbf{a}_i^{\text{dropout}} = \mathbf{a}_i \cdot m_i$$

Compute the loss $\mathcal{L}(\mathbf{Y}, \hat{\mathbf{Y}})$ where $\mathbf{Y}$ is the output of the network and $\hat{\mathbf{Y}}$ is the target output.

Calculate the gradients and Hessians of the loss with respect to the weights:

$$\frac{\partial \mathcal{L}}{\partial W_{ij}}, \quad \frac{\partial^2 \mathcal{L}}{\partial W_{ij}^2}$$

For each layer $L$ and loop $i$ within the layer, estimate the energy $E_i$ required for the current quanta size $l_i$:

$$E_i \leftarrow \text{DynAgent.estimateEnergy}(L, i, l_i)$$

If $E_i > E_b$, fuse tasks to reduce the overhead:

$$\text{FuseTasks}(L, i, l_i, E_b)$$

Update $E_i$ after task fusion:

$$E_i \leftarrow \text{DynAgent.estimateEnergy}(L, i, l_i)$$

Update the dropout mask $\mathbf{m}$ based on the sensitivities:

$$p_i = \frac{\beta \sum_j \frac{\partial^2 \mathcal{L}}{\partial W_{ij}^2} (W_{ij})^2}{\max\left(\sum_j \frac{\partial^2 \mathcal{L}}{\partial W_{ij}^2} (W_{ij})^2\right) + \epsilon}$$

$$m_i = \begin{cases} 0 & \text{if Bernoulli}(1 - p_i) = 0 \\ 1 & \text{otherwise} \end{cases}$$

Perform the backward pass to update the network weights, considering the dropout mask:

$$\mathbf{W} \leftarrow \mathbf{W} - \eta \frac{\partial \mathcal{L}}{\partial \mathbf{W}} \odot \mathbf{m}$$

where $\eta$ is the learning rate and $\odot$ denotes element-wise multiplication.

**Inference with Optimal Brain Damage Dropout and QuantaTask Optimization:** Check the available energy using DynAgent. If energy is below a threshold, increase the dropout rate to ensure the inference can be completed within the energy budget. Otherwise, maintain or reduce the dropout rate to improve accuracy. Perform the forward pass with the updated dropout mask to obtain the output $\mathbf{Y}$. This approach ensures that the network is robust to varying energy conditions by incorporating dynamic dropout influenced by the sensitivity of the weights, along with the QuantaTask optimization to handle energy constraints.

### D.3 Feature Map Reconstruction Error Dropout with QuantaTask Optimization

Feature Map Reconstruction Error Dropout leverages the reconstruction error of feature maps to adjust dropout rates, combined with the QuantaTask optimization to handle energy constraints in intermittent systems.

**Mathematical Formulation:** Let $\mathbf{W}$ be the weight matrix of a layer and $\mathbf{F}$ be the feature maps produced by the layer. The reconstruction error of a feature map $F_i$ is calculated as:

$$\text{RE}_i = \|\mathbf{F}_i - \hat{\mathbf{F}}_i\|_2$$

where $\hat{\mathbf{F}}_i$ is the reconstructed feature map, and $\|\cdot\|_2$ denotes the L2 norm.

Define the dropout probability $p_i$ for neuron $i$ based on the reconstruction error of its corresponding feature map. The idea is to use the reconstruction error to determine the probability:

$$p_i = \frac{\gamma \, \mathrm{RE}_i}{\max(\mathrm{RE}) + \epsilon}$$

where $\gamma$ is a scaling factor to adjust the overall dropout rate, and $\epsilon$ is a small constant to avoid division by zero.

Define a binary dropout mask $\mathbf{m} = [m_1, m_2, \ldots, m_n]$ where $m_i \in \{0, 1\}$. Each element of the mask is determined by sampling from a Bernoulli distribution with probability $1 - p_i$:

$$m_i \sim \mathrm{Bernoulli}(1 - p_i)$$

Apply the dropout mask during the forward pass. Let $\mathbf{a}_i$ denote the activation of neuron $i$:

$$\mathbf{a}_i^{\mathrm{dropout}} = \mathbf{a}_i \cdot m_i$$

**Training with Feature Map Reconstruction Error Dropout and QuantaTask Optimization:**
Initialize the network parameters $\mathbf{W}$, dropout mask $\mathbf{m}$, and scaling factor $\gamma$. Define the energy budget $E_b$ for a single quanta and for the entire inference. Initialize the loop iteration parameters $l$.

Compute the activations $\mathbf{a}$ and apply the dropout mask:

$$\mathbf{a}_i^{\mathrm{dropout}} = \mathbf{a}_i \cdot m_i$$

Compute the loss $\mathcal{L}(\mathbf{Y}, \hat{\mathbf{Y}})$ where $\mathbf{Y}$ is the output of the network and $\hat{\mathbf{Y}}$ is the target output.

Calculate the gradients of the loss with respect to the weights:

$$\frac{\partial \mathcal{L}}{\partial W_{ij}}$$

For each layer $L$ and loop $i$ within the layer, estimate the energy $E_i$ required for the current quanta size $l_i$:

$$E_i \leftarrow \mathrm{DynAgent.estimateEnergy}(L, i, l_i)$$

If $E_i > E_b$, fuse tasks to reduce the overhead:

$$\mathrm{FuseTasks}(L, i, l_i, E_b)$$

Update $E_i$ after task fusion:

$$E_i \leftarrow \mathrm{DynAgent.estimateEnergy}(L, i, l_i)$$

Update the dropout mask $\mathbf{m}$ based on the reconstruction error of the feature maps:

$$p_i = \frac{\gamma \, \mathrm{RE}_i}{\max(\mathrm{RE}) + \epsilon}$$

$$m_i = \begin{cases} 0 & \text{if } \mathrm{Bernoulli}(1 - p_i) = 0 \\ 1 & \text{otherwise} \end{cases}$$

Perform the backward pass to update the network weights, considering the dropout mask:

$$\mathbf{W} \leftarrow \mathbf{W} - \eta \frac{\partial \mathcal{L}}{\partial \mathbf{W}} \odot \mathbf{m}$$

where $\eta$ is the learning rate and $\odot$ denotes element-wise multiplication.

**Inference with Feature Map Reconstruction Error Dropout and QuantaTask Optimization:**
Check the available energy using DynAgent. If energy is below a threshold, increase the dropout rate to ensure the inference can be completed within the energy budget. Otherwise, maintain or reduce the dropout rate to improve accuracy. Perform the forward pass with the updated dropout mask to obtain the output $\mathbf{Y}$. This approach ensures that the network is robust to varying energy conditions by incorporating dynamic dropout influenced by the reconstruction error of the feature maps, along with the QuantaTask optimization to handle energy constraints.

## D.4 LEARNING SPARSE MASKS DROPOUT WITH QUANTATASK OPTIMIZATION

Learning Sparse Masks Dropout adapts dropout masks as learnable parameters within the network, inspired by Wen et al. (2016), combined with the QuantaTask optimization to handle energy constraints in intermittent systems.

**Mathematical Formulation:** Let $\mathbf{W}$ be the weight matrix of a layer. Define a binary dropout mask $\mathbf{m} = [m_1, m_2, \ldots, m_n]$ where $m_i \in \{0, 1\}$. In Learning Sparse Masks Dropout, the dropout masks are treated as learnable parameters. The mask values are determined using a sigmoid function to ensure they lie between 0 and 1:

$$m_i = \sigma(z_i)$$

where $z_i$ are learnable parameters and $\sigma(\cdot)$ is the sigmoid function.

Apply the dropout mask during the forward pass. Let $\mathbf{a}_i$ denote the activation of neuron $i$:

$$\mathbf{a}_i^{\text{dropout}} = \mathbf{a}_i \cdot m_i$$

Compute the loss $\mathcal{L}(\mathbf{Y}, \hat{\mathbf{Y}})$ where $\mathbf{Y}$ is the output of the network and $\hat{\mathbf{Y}}$ is the target output.

DynFit integrates closely with DynAgent, which serves as a repository of EH profiles and hardware characteristics. Let $\mathcal{Q}$ represent the set of execution quanta, where each quanta $q \in \mathcal{Q}$ is defined by a tuple $(l, e)$:

$$q = (l, e)$$

Here, $l$ is the number of loop iterations and $e$ is the estimated energy required for these iterations. The goal is to optimize the loop iteration parameter $l$ such that the energy consumption $E_q$ for each quanta $q$ is within the energy budget $E_b$:

$$\text{minimize} \quad \sum_{q \in \mathcal{Q}} E_q \quad \text{subject to} \quad E_q \leq E_b$$

**Training with Learning Sparse Masks Dropout and QuantaTask Optimization:** Initialize the network parameters $\mathbf{W}$, dropout mask parameters $\mathbf{z}$, and scaling factor $\alpha$. Define the energy budget $E_b$ for a single quanta and for the entire inference. Initialize the loop iteration parameters $l$.

Compute the activations $\mathbf{a}$ and apply the dropout mask:

$$m_i = \sigma(z_i)$$

$$\mathbf{a}_i^{\text{dropout}} = \mathbf{a}_i \cdot m_i$$

Compute the loss $\mathcal{L}(\mathbf{Y}, \hat{\mathbf{Y}})$. Calculate the gradients of the loss with respect to the weights and dropout mask parameters:

$$\frac{\partial \mathcal{L}}{\partial W_{ij}}, \quad \frac{\partial \mathcal{L}}{\partial z_i}$$

For each layer $L$ and loop $i$ within the layer, estimate the energy $E_i$ required for the current quanta size $l_i$:

$$E_i \leftarrow \text{DynAgent.estimateEnergy}(L, i, l_i)$$

If $E_i > E_b$, fuse tasks to reduce the overhead:

$$\text{FuseTasks}(L, i, l_i, E_b)$$

Update $E_i$ after task fusion:

$$E_i \leftarrow \text{DynAgent.estimateEnergy}(L, i, l_i)$$

Update the dropout mask parameters $\mathbf{z}$ based on the gradients:

$$z_i \leftarrow z_i - \eta \frac{\partial \mathcal{L}}{\partial z_i}$$

Perform the backward pass to update the network weights, considering the dropout mask:

$$\mathbf{W} \leftarrow \mathbf{W} - \eta \frac{\partial \mathcal{L}}{\partial \mathbf{W}} \odot \mathbf{m}$$

where $\eta$ is the learning rate and $\odot$ denotes element-wise multiplication.

**Inference with Learning Sparse Masks Dropout and QuantaTask Optimization:** Check the available energy using DynAgent. If energy is below a threshold, increase the dropout rate to ensure the inference can be completed within the energy budget. Otherwise, maintain or reduce the dropout rate to improve accuracy. Perform the forward pass with the updated dropout mask to obtain the output $\mathbf{Y}$. This approach ensures that the network is robust to varying energy conditions by incorporating dynamic dropout with learnable mask parameters, along with the QuantaTask optimization to handle energy constraints.

### D.5 NEURON SHAPLEY VALUE DROPOUT WITH QUANTATASK OPTIMIZATION

Neuron Shapley Value Dropout applies the concept of Shapley values from game theory (Aas et al., 2021) to assess neuron importance for dropout, combined with the QuantaTask optimization to handle energy constraints in intermittent systems.

**Mathematical Formulation:** The Shapley value $\phi_i$ of neuron $i$ is a measure of its contribution to the overall network performance. It is calculated by considering all possible subsets of neurons and computing the marginal contribution of neuron $i$ to the network's output:

$$\phi_i = \frac{1}{|\mathcal{N}|!} \sum_{S \subseteq \mathcal{N} \setminus \{i\}} \frac{|S|!(|\mathcal{N}| - |S| - 1)!}{|\mathcal{N}|} [\mathcal{L}(S \cup \{i\}) - \mathcal{L}(S)]$$

where $\mathcal{N}$ is the set of all neurons, $S$ is a subset of neurons not containing $i$, and $\mathcal{L}(\cdot)$ denotes the loss function.

Define the dropout probability $p_i$ for neuron $i$ based on its Shapley value. Neurons with lower Shapley values are more likely to be dropped:

$$p_i = \frac{\delta}{\phi_i + \epsilon}$$

where $\delta$ is a scaling factor to adjust the overall dropout rate, and $\epsilon$ is a small constant to avoid division by zero.

Define a binary dropout mask $\mathbf{m} = [m_1, m_2, \ldots, m_n]$ where $m_i \in \{0, 1\}$. Each element of the mask is determined by sampling from a Bernoulli distribution with probability $1 - p_i$:

$$m_i \sim \text{Bernoulli}(1 - p_i)$$

Apply the dropout mask during the forward pass. Let $\mathbf{a}_i$ denote the activation of neuron $i$:

$$\mathbf{a}_i^{\text{dropout}} = \mathbf{a}_i \cdot m_i$$

**Training with Neuron Shapley Value Dropout and QuantaTask Optimization:** Initialize the network parameters $\mathbf{W}$, dropout mask $\mathbf{m}$, and scaling factor $\delta$. Define the energy budget $E_b$ for a single quanta and for the entire inference. Initialize the loop iteration parameters $l$.

Compute the activations $\mathbf{a}$ and apply the dropout mask:

$$\mathbf{a}_i^{\text{dropout}} = \mathbf{a}_i \cdot m_i$$

Compute the loss $\mathcal{L}(\mathbf{Y}, \hat{\mathbf{Y}})$ where $\mathbf{Y}$ is the output of the network and $\hat{\mathbf{Y}}$ is the target output.

Calculate the Shapley values $\phi_i$ for each neuron based on their contribution to the network's performance.

For each layer $L$ and loop $i$ within the layer, estimate the energy $E_i$ required for the current quanta size $l_i$:

$$E_i \leftarrow \text{DynAgent.estimateEnergy}(L, i, l_i)$$

If $E_i > E_b$, fuse tasks to reduce the overhead:

$$\text{FuseTasks}(L, i, l_i, E_b)$$

Update $E_i$ after task fusion:

$$E_i \leftarrow \text{DynAgent.estimateEnergy}(L, i, l_i)$$

Update the dropout mask $\mathbf{m}$ based on the Shapley values:

$$p_i = \frac{\delta}{\phi_i + \epsilon}$$

$$m_i = \begin{cases} 0 & \text{if Bernoulli}(1 - p_i) = 0 \\ 1 & \text{otherwise} \end{cases}$$

Perform the backward pass to update the network weights, considering the dropout mask:

$$\mathbf{W} \leftarrow \mathbf{W} - \eta \frac{\partial \mathcal{L}}{\partial \mathbf{W}} \odot \mathbf{m}$$

where $\eta$ is the learning rate and $\odot$ denotes element-wise multiplication.

**Inference with Neuron Shapley Value Dropout and QuantaTask Optimization:** Check the available energy using DynAgent. If energy is below a threshold, increase the dropout rate to ensure the inference can be completed within the energy budget. Otherwise, maintain or reduce the dropout rate to improve accuracy. Perform the forward pass with the updated dropout mask to obtain the output $\mathbf{Y}$. This approach ensures that the network is robust to varying energy conditions by incorporating dynamic dropout influenced by the Shapley values of the neurons, along with the QuantaTask optimization to handle energy constraints.

### D.6   TAYLOR EXPANSION DROPOUT WITH QUANTATASK OPTIMIZATION

Taylor Expansion Dropout uses Taylor expansion (Li et al., 2016) to evaluate the impact of neurons on loss for dropout adjustments, combined with the QuantaTask optimization to handle energy constraints in intermittent systems.

**Mathematical Formulation:** Let $\mathbf{W}$ be the weight matrix of a layer. The impact of neuron $i$ on the loss function $\mathcal{L}$ can be approximated using the first-order Taylor expansion:

$$\Delta \mathcal{L}_i \approx \left| \frac{\partial \mathcal{L}}{\partial \mathbf{a}_i} \mathbf{a}_i \right|$$

where $\mathbf{a}_i$ is the activation of neuron $i$, and $\frac{\partial \mathcal{L}}{\partial \mathbf{a}_i}$ is the gradient of the loss with respect to the activation.

Define the dropout probability $p_i$ for neuron $i$ based on the Taylor expansion approximation of its impact on the loss:

$$p_i = \frac{\lambda}{\left| \frac{\partial \mathcal{L}}{\partial \mathbf{a}_i} \mathbf{a}_i \right| + \epsilon}$$

where $\lambda$ is a scaling factor to adjust the overall dropout rate, and $\epsilon$ is a small constant to avoid division by zero.

Define a binary dropout mask $\mathbf{m} = [m_1, m_2, \ldots, m_n]$ where $m_i \in \{0, 1\}$. Each element of the mask is determined by sampling from a Bernoulli distribution with probability $1 - p_i$:

$$m_i \sim \text{Bernoulli}(1 - p_i)$$

Apply the dropout mask during the forward pass. Let $\mathbf{a}_i$ denote the activation of neuron $i$:

$$\mathbf{a}_i^{\text{dropout}} = \mathbf{a}_i \cdot m_i$$

**Training with Taylor Expansion Dropout and QuantaTask Optimization:** Initialize the network parameters $\mathbf{W}$, dropout mask $\mathbf{m}$, and scaling factor $\lambda$. Define the energy budget $E_b$ for a single quanta and for the entire inference. Initialize the loop iteration parameters $l$.

Compute the activations $\mathbf{a}$ and apply the dropout mask:

$$\mathbf{a}_i^{\text{dropout}} = \mathbf{a}_i \cdot m_i$$

Compute the loss $\mathcal{L}(\mathbf{Y}, \hat{\mathbf{Y}})$ where $\mathbf{Y}$ is the output of the network and $\hat{\mathbf{Y}}$ is the target output.

Calculate the gradients of the loss with respect to the activations:

$$\frac{\partial \mathcal{L}}{\partial \mathbf{a}_i}$$

For each layer $L$ and loop $i$ within the layer, estimate the energy $E_i$ required for the current quanta size $l_i$:

$$E_i \leftarrow \text{DynAgent.estimateEnergy}(L, i, l_i)$$

If $E_i > E_b$, fuse tasks to reduce the overhead:

$$\text{FuseTasks}(L, i, l_i, E_b)$$

Update $E_i$ after task fusion:

$$E_i \leftarrow \text{DynAgent.estimateEnergy}(L, i, l_i)$$

Update the dropout mask $\mathbf{m}$ based on the Taylor expansion approximation:

$$p_i = \frac{\lambda}{\left| \frac{\partial \mathcal{L}}{\partial \mathbf{a}_i} \mathbf{a}_i \right| + \epsilon}$$

$$m_i = \begin{cases} 0 & \text{if } \text{Bernoulli}(1 - p_i) = 0 \\ 1 & \text{otherwise} \end{cases}$$

Perform the backward pass to update the network weights, considering the dropout mask:

$$\mathbf{W} \leftarrow \mathbf{W} - \eta \frac{\partial \mathcal{L}}{\partial \mathbf{W}} \odot \mathbf{m}$$

where $\eta$ is the learning rate and $\odot$ denotes element-wise multiplication.

**Inference with Taylor Expansion Dropout and QuantaTask Optimization:** Check the available energy using DynAgent. If energy is below a threshold, increase the dropout rate to ensure the inference can be completed within the energy budget. Otherwise, maintain or reduce the dropout rate to improve accuracy. Perform the forward pass with the updated dropout mask to obtain the output $\mathbf{Y}$. This approach ensures that the network is robust to varying energy conditions by incorporating dynamic dropout influenced by the Taylor expansion approximation of the neurons' impact on the loss, along with the QuantaTask optimization to handle energy constraints.

## E    WORKINGS OF RE-RAM CROSSBAR

### E.1    RE-RAM CROSS-BAR FOR DNN INFERENCE:

ReRAM x-bars are an emerging class of computing devices that leverage resistive random-access memory (ReRAM) technology for efficient and low-power computing. These devices can perform multiplication and addition operations in a single operation, making them ideal for many signal processing and machine learning applications. Moreover, these devices can also be used for performing convolution operations, which are widely used in image and signal processing applications.

#### E.1.1    SIMPLE SINGLE CELL EXAMPLE:

consider a simple example of a ReRAM crossbar array with two cells, where V1 and V2 are the input voltages, G1 and G2 are the conductance values of the ReRAM devices, and I1 and I2 are the resulting output currents. To perform multiplication-addition, we first apply the input voltages V1 and V2 to the rows of the crossbar array. The conductance values G1 and G2 of the ReRAM devices are set to the corresponding weight values for the multiplication operation. The output currents I1 and I2 are then computed as follows:

$$I = I1 + I2$$
$$= G1 \times V1 + G2 \times V2$$

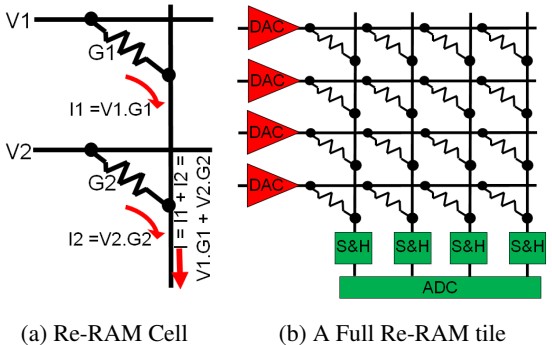

(a) Re-RAM Cell        (b) A Full Re-RAM tile

Figure 5: DNN computation using ReRAM xBAR.

Here, the output currents I1 and I2 are the result of the multiplication of the input voltages V1 and V2 by their respective weight values, which are summed together using the crossbar wires. Please refer to Figure 5a for more details. As we can see, the input voltages V1 and V2 are applied to the rows of the crossbar array, while the conductance values G1 and G2 are applied to the columns. The output currents I1 and I2 are the result of the multiplication-addition operation, and are obtained by summing the currents flowing through the ReRAM devices.

In practice, ReRAM crossbar arrays can have many more cells, and can be used to perform more complex multiplication-addition and convolution operations. However, the basic principle remains the same, where the input signals are applied to the rows, the weights are applied to the columns, and the output signals are obtained by summing the currents flowing through the ReRAM devices.

### E.1.2 EXTENDING TO COMPLEX COMPUTE:

In order to perform multiplication-addition in ReRAM x-bars, two arrays of weights and inputs are used. The inputs are fed to the x-bar, which is a two-dimensional array of ReRAM crossbar arrays. The crossbar arrays are composed of a set of row and column wires that intersect at a set of ReRAM devices (refer Figure 5b). The ReRAM devices are programmed to have different resistance values, which are used to store the weights.

During the multiplication-addition operation, the input signals are applied to the rows of the x-bar, and the weights are applied to the columns. The output of each ReRAM device is the product of the input and weight signals, which are added together using the crossbar wires. This results in a single output signal that represents the sum of the weighted inputs.

To perform convolution, ReRAM x-bars use a similar approach, but with a more complex circuit. The input signal is applied to the x-bar in the same way, but the weights are now applied in a more structured way. Specifically, the weights are arranged in a way that mimics the convolution operation, such that each weight corresponds to a specific location in the input signal. To perform the convolution operation, the input signal is applied to the rows of the x-bar, and the weights are applied to the columns in a structured way. The output signal is obtained by summing the weighted input signals over a sliding window, which moves across the input signal to compute the convolution.

At the circuit level, the ReRAM x-bar for multiplication-addition typically includes several components, such as digital-to-analog converters (DACs), analog-to-digital converters (ADCs), shift registers, and hold capacitors. The DACs and ADCs are used to convert the digital input and weight signals into analog signals that can be applied to the rows and columns of the x-bar. The shift registers are used to apply the weight signals in a structured way, and the hold capacitors are used to store the analog signals during the multiplication-addition operation. Similarly, for performing convolution, the ReRAM x-bar typically includes additional components, such as delay lines and adders. The delay lines are used to implement the sliding window for the convolution operation, while the adders are used to sum the weighted input signals over the sliding window.

# F PSEUDO CODES

## F.1 DEPTH-WISE SEPARABLE CONVOLUTION 2D USING TI LEA

Depth-wise separable convolution is an efficient form of convolution that reduces the computational cost compared to standard convolution. Here we describe the implementation of depth-wise separable convolution 2D using the Low Energy Accelerator (LEA) in Texas Instruments' MSP430 microcontrollers.

### F.1.1 DEPTH-WISE SEPARABLE CONVOLUTION 2D USING CONV1D

The pseudo code described in Algorithm 1 implements a depth-wise separable convolution 2D (DWSConv2D) using a 1D convolution primitive function (conv1D). The DWSConv2D function takes four inputs: an input matrix, depth-wise kernels (DWsKernels), point-wise kernels (PtWsKernel), and an output matrix. The depth-wise separable convolution is performed in two main steps: depth-wise convolution and point-wise convolution.

---

**Algorithm 1** Implementing Depth-wise Separable Convolution - DWSConv2D() using CONV1D ()

1: **Function** DWSepConv2D($inputMatrix$, $DWsKernels$, $PtWsKernel$, $outputMatrix$):
2:     Initialize $DWsOutput$ with zero values, same shape as $inputMatrix$
3:     # Depth-wise Separable (DWs) convolution
4:     **for** $c \leftarrow 0$ **to** $channels(inputMatrix) - 1$:
5:         # Apply 1D convolution along rows
6:         **for** $i \leftarrow 0$ **to** $rows(inputMatrix[c]) - 1$:
7:             conv1D($inputMatrix[c][i, :]$, $DWsKernels[c][0, :]$, $DWsOutput[c][i, :]$)
8:         # Apply 1D convolution along columns
9:         **for** $j \leftarrow 0$ **to** $cols(DWsOutput[c]) - 1$:
10:             conv1D($DWsOutput[c][:, j]$, $DWsKernels[c][:, 0]$, $DWsOutput[c][:, j]$)
11:     # Point-wise (PtWs) convolution
12:     Initialize $finalOutput$ with zero values, with shape [rows($DWsOutput$), cols($DWsOutput$), channels($PtWsKernel$)]
13:     **for** $i \leftarrow 0$ **to** $rows(DWsOutput) - 1$:
14:         **for** $j \leftarrow 0$ **to** $cols(DWsOutput) - 1$:
15:             **for** $k \leftarrow 0$ **to** $channels(PtWsKernel) - 1$:
16:                 Initialize $PtWsSum \leftarrow 0$
17:                 **for** $c \leftarrow 0$ **to** $channels(DWsOutput) - 1$:
18:                     $PtWsSum \leftarrow PtWsSum + DWsOutput[c][i][j] \times PtWsKernel[c][k]$
19:                 $finalOutput[i][j][k] \leftarrow PtWsSum$
20:     **return** $finalOutput$

---

### F.1.2 PSEUDOCODE WITH MICRO-CONTROLLER PRIMITIVES

The following pseudocode describes the steps to implement depth-wise separable convolution using LEA primitives from TI's DSP Library.

---

**Algorithm 2** depth-wise Separable Convolution 2D Using TI LEA

---

1: **function** DWSEPCONV2D($inputMatrix$, $DWsKernels$, $PtWsKernel$, $outputMatrix$)
2:     Initialize $tempMatrix1$ and $tempMatrix2$ with zero values, same shape as $inputMatrix$
3:     // Depth-wise convolution
4:     **for** $c \leftarrow 0$ **to** $channels(inputMatrix) - 1$ **do**
5:         // Apply 1D convolution along rows
6:         **for** $i \leftarrow 0$ **to** $rows(inputMatrix[c]) - 1$ **do**
7:             MSP_CONV_IQ31($inputMatrix[c][i,:]$,             $DWsKernels[c][0,:]$, $tempMatrix1[c][i,:]$, $cols(inputMatrix)$, $FILTER\_SIZE$)
8:         **end for**
9:         // Apply 1D convolution along columns
10:         **for** $j \leftarrow 0$ **to** $cols(tempMatrix1[c]) - 1$ **do**
11:             MSP_CONV_IQ31($tempMatrix1[c][:,j]$,             $DWsKernels[c][:,0]$, $tempMatrix2[c][:,j]$, $rows(tempMatrix1)$, $FILTER\_SIZE$)
12:         **end for**
13:     **end for**
14:     // Point-wise convolution
15:     Initialize $finalOutput$ with zero values, shape [rows($tempMatrix2$), cols($tempMatrix2$), channels($PtWsKernel$)]
16:     **for** $i \leftarrow 0$ **to** $rows(tempMatrix2) - 1$ **do**
17:         **for** $j \leftarrow 0$ **to** $cols(tempMatrix2) - 1$ **do**
18:             **for** $k \leftarrow 0$ **to** $channels(PtWsKernel) - 1$ **do**
19:                 Initialize $PtWsSum \leftarrow 0$
20:                 **for** $c \leftarrow 0$ **to** $channels(tempMatrix2) - 1$ **do**
21:                     $PtWsSum \leftarrow PtWsSum + tempMatrix2[c][i][j] \times PtWsKernel[c][k]$
22:                 **end for**
23:                 $finalOutput[i][j][k] \leftarrow PtWsSum$
24:             **end for**
25:         **end for**
26:     **end for**
27:     **return** $finalOutput$
28: **end function**

---

### F.1.3 IMPLEMENTATION CODE

C code that implements the pseudo-code using TI's LEA (Instruments, 2024b) functions.

```
#include <msp430.h>
#include "DSPLib.h"

#define ROWS 64
#define COLS 64
#define CHANNELS 3
#define FILTER_SIZE 3

// Initialize your input, depth-wise kernels, point-wise kernels,
// and output matrices appropriately
_q31 inputMatrix[CHANNELS][ROWS][COLS];
_q31 DWsKernels[CHANNELS][FILTER_SIZE][FILTER_SIZE];
_q31 PtWsKernel[CHANNELS][CHANNELS];
_q31 tempMatrix1[CHANNELS][ROWS][COLS];
_q31 tempMatrix2[CHANNELS][ROWS][COLS];
_q31 finalOutput[ROWS][COLS][CHANNELS];

void DWSepConv2D() {
    // Depth-wise convolution
    for (int c = 0; c < CHANNELS; c++) {
        // Apply 1D convolution along rows
```

```
        for (int i = 0; i < ROWS; i++) {
            msp_conv_iq31(&inputMatrix[c][i][0], DWsKernels[c][0],
            &tempMatrix1[c][i][0], COLS, FILTER_SIZE);
        }
        // Apply 1D convolution along columns
        for (int j = 0; j < COLS; j++) {
            msp_conv_iq31(&tempMatrix1[c][0][j], DWsKernels[c][0],
            &tempMatrix2[c][0][j], ROWS, FILTER_SIZE);
        }
    }

    // Point-wise convolution
    for (int i = 0; i < ROWS; i++) {
        for (int j = 0; j < COLS; j++) {
            for (int k = 0; k < CHANNELS; k++) {
                _q31 PtWsSum = 0;
                for (int c = 0; c < CHANNELS; c++) {
                    PtWsSum += tempMatrix2[c][i][j] * PtWsKernel[c][k];
                }
                finalOutput[i][j][k] = PtWsSum;
            }
        }
    }
}
```

## F.2   TASK-BASED CONV2D

Here we describe the implementation of a task-based 'CONV2D' function using the Low Energy
Accelerator (LEA) in Texas Instruments' MSP430 microcontrollers. The function is designed to
handle energy constraints by decomposing the convolution loops into smaller quanta tasks. Foloowing
are the outline of the requirements:

1. Define 'QuantaTask' as the minimum iterations that can run.

2. Decomposable loops: Each 'QuantaTask' runs a certain part of the loop.

3. Check for sufficient energy before launching a 'QuantaTask'.

4. Fuse multiple 'QuantaTask's to minimize load/store operations.

5. Check for power loss after each 'QuantaTask' or fused 'QuantaTask' and checkpoint if
   necessary.

---

**Algorithm 3** Task-Based CONV2D Using TI LEA

---

1: **Define** $QuantaTask$ as the minimum iterations we can run
2: **function** TASKBASEDCONV2D($inputMatrix$, $kernel$, $outputMatrix$)
3:     Initialize $tempMatrix$ with zero values, same shape as $inputMatrix$
4:     $rows \leftarrow$ rows of $inputMatrix$
5:     $cols \leftarrow$ cols of $inputMatrix$
6:     $kernelSize \leftarrow$ size of $kernel$
7:     $i \leftarrow 0$
8:     **while** $i < rows$ **do**
9:         $j \leftarrow 0$
10:        **while** $j < cols$ **do**
11:           $remainingEnergy \leftarrow$ CHECKENERGY($QuantaTask$)
12:           **if** $remainingEnergy$ is sufficient **then**
13:              EXECUTEQUANTATASK($i, j, inputMatrix, kernel, tempMatrix$)
14:              UPDATEPROGRESS($i, j, QuantaTask$)
15:              **if** POWERLOSSDETECTED **then**
16:                 CHECKPOINT($i, j, tempMatrix$)
17:                 **break**
18:              **end if**
19:           **else**
20:              **wait for energy to replenish**
21:           **end if**
22:        **end while**
23:     **end while**
24:     FUSETASKS
25:     **return** $outputMatrix$
26: **end function**
27: **function** EXECUTEQUANTATASK($i, j, inputMatrix, kernel, tempMatrix$)
28:     **for** $ki \leftarrow 0$ **to** $kernelSize - 1$ **do**
29:         **for** $kj \leftarrow 0$ **to** $kernelSize - 1$ **do**
30:           MSP_CONV_IQ31($inputMatrix[i + ki][j + kj]$, $kernel[ki][kj]$, $tempMatrix[i][j]$, $cols$, $kernelSize$)
31:         **end for**
32:     **end for**
33: **end function**
34: **function** FUSETASKS
35:     $remainingEnergy \leftarrow$ CHECKENERGY($multiple\_QuantaTask$)
36:     **while** $remainingEnergy$ is sufficient **do**
37:         EXECUTEQUANTATASK($i, j, inputMatrix, kernel, tempMatrix$)
38:         UPDATEPROGRESS($i, j, multiple\_QuantaTask$)
39:         $remainingEnergy \leftarrow$ CHECKENERGY($multiple\_QuantaTask$)
40:         **if** POWERLOSSDETECTED **then**
41:           CHECKPOINT($i, j, tempMatrix$) **break**
42:         **end if**
43:     **end while**
44: **end function**
45: **function** CHECKENERGY($QuantaTask$)
46:     # Check if there is enough energy to run the quanta task
47:     **return** $remainingEnergy$
48: **end function**
49: **function** POWERLOSSDETECTED
50:     # Check if power loss is detected
51:     **return** $powerLoss$
52: **end function**
53: **function** CHECKPOINT($i, j, tempMatrix$)
54:     # Save the current state to non-volatile memory
55: **end function**
56: **function** UPDATEPROGRESS($i, j, QuantaTask$)
57:     # Update loop indices based on the quanta task executed
58:     $j \leftarrow j + QuantaTask$
59:     **if** $j \geq cols$ **then**
60:         $j \leftarrow 0$
61:         $i \leftarrow i + QuantaTask$
62:     **end if**
63: **end function**

---

### F.2.1 IMPLEMENTATION CODE

```c
#include <msp430.h>
#include "DSPLib.h"

#define ROWS 64
#define COLS 64
#define KERNEL_SIZE 3
#define QuantaTask 8

// Define the FeRAM addresses for storing the checkpoint data
#define FERAM_ADDR_I 0xF000
#define FERAM_ADDR_J 0xF002
#define FERAM_ADDR_TEMPMATRIX 0xF004

_q31 inputMatrix[ROWS][COLS];
_q31 kernel[KERNEL_SIZE][KERNEL_SIZE];
_q31 tempMatrix[ROWS][COLS];
_q31 outputMatrix[ROWS][COLS];

void TaskBasedCONV2D() {
    int rows = ROWS;
    int cols = COLS;
    int kernelSize = KERNEL_SIZE;
    int i = 0;

    while (i < rows) {
        int j = 0;
        while (j < cols) {
            int remainingEnergy = CheckEnergy(QuantaTask);
            if (remainingEnergy > 0) {
                ExecuteQuantaTask(i, j, inputMatrix, kernel, tempMatrix);
                UpdateProgress(&i, &j, QuantaTask);
                if (PowerLossDetected()) {
                    Checkpoint(i, j, tempMatrix);
                    break;
                }
            } else {
                // Wait for energy to replenish
            }
        }
    }

    FuseTasks();
}

void ExecuteQuantaTask(int i, int j, _q31 inputMatrix[][COLS],
    _q31 kernel[][KERNEL_SIZE], _q31 tempMatrix[][COLS]) {
    for (int ki = 0; ki < KERNEL_SIZE; ki++) {
        for (int kj = 0; kj < KERNEL_SIZE; kj++) {
            msp_conv_iq31(&inputMatrix[i + ki][j + kj],
                &kernel[ki][kj], &tempMatrix[i][j], COLS, KERNEL_SIZE);
        }
    }
}

void FuseTasks() {
    int remainingEnergy = CheckEnergy(QuantaTask);
    while (remainingEnergy > 0) {
```

```
        ExecuteQuantaTask(i, j, inputMatrix, kernel, tempMatrix);
        UpdateProgress(&i, &j, QuantaTask);
        remainingEnergy = CheckEnergy(QuantaTask);
        if (PowerLossDetected()) {
            Checkpoint(i, j, tempMatrix);
            break;
        }
    }
}

int CheckEnergy(int QuantaTask) {
    // Energy checking  - HW interrupt
    return 1;
}

int PowerLossDetected() {
    // ower loss detection - HW interrupt logic
    return 0;
}

void Checkpoint(int i, int j, _q31 tempMatrix[][COLS]) {
    // Disable interrupts to prevent corruption during the write process
    __disable_interrupt();

    // Save the indices i and j to FeRAM
    *((volatile int*)FERAM_ADDR_I) = i;
    *((volatile int*)FERAM_ADDR_J) = j;

    // Save the current state of tempMatrix to FeRAM
    // Assuming tempMatrix is a 2D array of dimensions [ROWS][COLS]
    for (int row = 0; row < ROWS; row++) {
        for (int col = 0; col < COLS; col++) {
            ((volatile _q31*)FERAM_ADDR_TEMPMATRIX)[row * COLS + col]
                = tempMatrix[row][col];
        }
    }

    // Re-enable interrupts
    __enable_interrupt();
}

void RestoreCheckpoint(int *i, int *j, _q31 tempMatrix[][COLS]) {
    // Disable interrupts
    __disable_interrupt();

    // Restore the indices i and j from FeRAM
    *i = *((volatile int*)FERAM_ADDR_I);
    *j = *((volatile int*)FERAM_ADDR_J);

    // Restore the state of tempMatrix from FeRAM
    for (int row = 0; row < ROWS; row++) {
        for (int col = 0; col < COLS; col++) {
            tempMatrix[row][col] = ((volatile _q31*)
                FERAM_ADDR_TEMPMATRIX)[row * COLS + col];
        }
    }

    // Re-enable interrupts
    __enable_interrupt();
```

```
}

void UpdateProgress(int *i, int *j, int QuantaTask) {
    *j += QuantaTask;
    if (*j >= COLS) {
        *j = 0;
        *i += QuantaTask;
    }
}
```

