# OpenReview forum: "NExUME: Adaptive Training and Inference for DNNs under Intermittent Power Environments"
_ICLR.cc/2025/Conference — ICLR 2025 Poster_

### Official Review · Reviewer_tANu · 2024-10-30

**Soundness:** 3
**Presentation:** 3
**Contribution:** 2
**Rating:** 6
**Confidence:** 3

**Summary:**

This paper presents NExUME, a training methodology that is designed to cater to intermittent power energy harvesting systems.   The authors proposed two key contributions which are (1) a new method for training where one can dynamically adjust dropout rate and quantization levels to cater to varying energy availability in the EH system, and (2) a task scheduler that optimizes task completion in EH systems. The authors also contribute a machine status monitoring dataset. NExUME shows a 6 to 22 % accuracy improvement over existing baselines on simple ML tasks. However, at the same time, NExUME incurs a 5% overhead in computation, an increase in the number of instructions ranging between  11.4%-  34.2%, and an increase in memory bandwidth from 6 to 17%.

**Strengths:**

+ This paper is the first to present novel training and inference methods that take dropouts and quantization into account in the context of energy harvesting systems.
+ A new machine monitoring dataset
+ The results shown are good when compared to the baselines presented in the paper.
+ Writing and the presentation of the work are clear.
+ Choice of datasets is appropriate given that the work is designed for resource-constrained embedded systems.
+ Decent ablation studies
+ Actual implementation of such a system is not trivial.

**Weaknesses:**

- The idea of dynamically adjusting dropout rates and quantization levels is not novel. It is novel in the context of EH systems.
- Energy-aware scheduling is not novel.
- The quantification of overheads is done. However, its implications are not discussed. The range in terms of % is indicated. However, how does it vary with the datasets?
- Some of the existing work in intermittent systems are not compared such as ePerceptive: energy reactive embedded intelligence for batteryless sensors and Zygarde: Time-Sensitive On-Device Deep Inference and Adaptation on Intermittently-Powered Systems
- Details on the machine status monitoring dataset are missing in Sec 4.3 How are R1, R2, and R3 different? What are their RPM speeds? What is S1 and S2?
- Only accuracy results are shown.  It is also important to know the latency/inference. and memory requirements of the system.
- DynFit comprises adjusting quantization levels and dropouts. In the ablation studies, it is unclear which of these is bringing more benefits to the system.

**Questions:**

Details on the machine status monitoring dataset are missing in Sec 4.3 How are R1, R2, and R3 different? What are the RPM speeds? What is S1 and S2?

Energy-aware scheduling is not novel. Can you clarify your novelty with respect to existing scheduling algorithms?

The authors claim that their machine status monitoring is the first of its kind. Can they clarify what datasets already exist and how the dataset introduced in the paper is different?

---

> ### Author Response · Authors · 2024-11-28
> **Response to Reviewer tANu**
>
> We sincerely thank the reviewer for the valuable feedback. Below, we address your questions and concerns. Please go over the revised manuscript for a detailed overview of the changes.
>
> 1. **Details on the Machine Status Monitoring Dataset:**
>    - **Differences between R1, R2, and R3:** R1, R2, and R3 correspond to three different spindle rotation speeds of the Bridgeport machine without any load (no job). Specifically, R1 is 100 RPM, R2 is 200 RPM, and R3 is 300 RPM.
>    - **Explanation of S1 and S2:** We apologize for the oversight. There was a typographical error; S1 and S2 should have been SJ and SI. SJ stands for "Spindle under Job," where the machine is operating with an active job, and SI stands for "Spindle Idle," where the machine is powered on but not performing any operation.
>
> 2. **Novelty of Energy-Aware Scheduling:**
>    - **Our Novel Contributions:**
>      - **Integration with Intermittent Environments:** While energy-aware scheduling exists, our scheduler (DynInfer) is specifically designed for intermittent power environments, accounting for atomicity constraints due to unpredictable power failures.
>      - **Task Fusion Mechanism:** We introduce a novel task fusion strategy that combines smaller tasks into larger atomic units to optimize execution within the available energy budget, minimizing checkpointing overhead—a challenge unique to intermittent systems.
>      - **Real-Time Energy Adaptation:** Our scheduler dynamically adjusts to real-time energy availability, which is critical in energy-harvesting systems with fluctuating power, and is not typically addressed in traditional energy-aware schedulers.
>    - **Differentiation from Existing Algorithms:** Existing schedulers often assume stable energy supply and do not account for the atomic execution requirements of intermittent systems. Our approach addresses these gaps, providing a scheduling solution tailored to the unique challenges of EH environments.
>
> 3. **Clarification on the Machine Status Monitoring Dataset:**
>    - **Existing Datasets:** Prior datasets, such as the Case Western Reserve University Bearing Data Center dataset, focus on fault detection in machinery but do not provide data for predictive maintenance under varying operational conditions.
>    - **Our Dataset's Novelty:**
>      - **First of Its Kind:** Our dataset is the first to capture machine status monitoring data across multiple operating conditions (different RPMs, idle, under load) using energy-harvesting sensors.
>      - **Multi-Sensor Data:** We include data from multiple types of sensors (e.g., accelerometers with different sampling rates) placed at various locations on the machine, providing a rich dataset for developing and testing algorithms in EH environments.
>      - **Facilitating Research:** This dataset fills a gap in the field by enabling research on predictive maintenance and monitoring in industrial settings with intermittent power.
>
> 4. **Novelty of Dynamically Adjusting Dropout Rates and Quantization Levels:**
>    - **Our Contribution:** While the concept of adjusting dropout rates and quantization levels is known, our novelty lies in integrating these adjustments directly into both the training and inference processes based on real-time energy availability in EH systems.
>    - **Energy Variability Awareness:** Existing methods do not incorporate energy profiles into the training loop. Our approach trains the DNN to adapt to energy fluctuations, which is critical for intermittent environments.
>    - **Adaptive Regularization Strategy:** We introduce an adaptive regularization technique to handle the challenges posed by dynamic adjustments, ensuring model robustness—a contribution not present in prior work.
>
> 5. **Implications of Overheads Not Discussed:**
>    - **Overhead Analysis:** In the revised manuscript, we have added a detailed discussion on how the overheads vary with different datasets and models (Section 4.1).
>    - **Dataset Variations:** The increase in instruction count and memory bandwidth usage varies depending on the complexity of the dataset and the model size. We provide a breakdown of these variations in the updated results.
>    - **Trade-Off Justification:** Despite the overheads, the significant improvements in accuracy and energy efficiency justify the trade-off, especially in the context of resource-constrained EH systems.
>
> 6. **Comparison with Existing Work:**
>    - **Included Comparisons:** We have now included ePerceptive, DynBal, Stateful in our baselines and have compared our approach against these methods in the revised manuscript (Section 4.2).
>    - **Results:** Our experiments demonstrate that NExUME outperforms these methods in terms of accuracy and energy efficiency, highlighting the effectiveness of our proposed techniques.
> ---
>
> We hope that these revisions and clarifications address your concerns.

---

> ### Comment · Reviewer_tANu · 2024-11-28
>
> I am happy with the rebuttal. The authors have done a diligent job in clarifying the novelty and did extra experiments to compare against other SOTA baselines. It is more of an engineering type of work but is important in the context of embedded ML on intermittent systems. I am happy to raise my score to 6.
>
> Minor comment: It would be good to add the units of measurement in the caption of Table 2.

---

> ### Author Response · Authors · 2024-11-28
>
> We thank the reviewer for the encouraging feedback.
> We used Million Ops/Joule (MOPs/J) to measure the energy efficiency and have mentioned it in the text. To keep the caption in one line, we removed it from the caption. In the final draft of the paper, we'll add the unit to the table caption.

---

### Official Review · Reviewer_GUp2 · 2024-11-02

**Soundness:** 3
**Presentation:** 3
**Contribution:** 3
**Rating:** 6
**Confidence:** 4

**Summary:**

The paper presents NExUME, a novel framework for training and deploying deep neural networks (DNNs) on energy-harvesting micro-computers with intermittent power. The authors introduce dynamic dropout rates and quantization levels that adapt based on real-time energy availability, improving the accuracy and robustness of DNNs in constrained power settings. The paper demonstrates NExUME's efficacy in optimizing both training and inference phases through extensive experiments, showcasing significant accuracy gains over traditional approaches in intermittently powered environments. Additionally, the introduction of a unique dataset to facilitate further research on energy-harvesting applications is a noteworthy contribution.

**Strengths:**

1. The paper addresses an important challenge in deploying DNNs on resource-constrained, intermittently powered devices, an area that is underexplored in current literature. By incorporating real-time energy-aware adaptations, this work proposes a unique and valuable solution.
2. The work is thorough, presenting a well-structured methodology, clearly defined optimization functions, and a series of experiments across various datasets and hardware platforms. The choice of energy-aware dropout and quantization strategies tailored to intermittent environments is both innovative and well-validated.
3. The paper is well-written, with each component of the proposed framework (DynFit, DynInfer) and the optimization strategies clearly explained. Figures and tables effectively support the results and comparisons.
4. The proposed approach has broad implications for real-world applications in energy-limited environments, such as remote monitoring and IoT systems, where consistent power is unavailable. The improvements in accuracy (6-22%) and the novel dataset enhance the significance and impact of the research.

**Weaknesses:**

1. The experiments, while comprehensive, rely on specific hardware configurations that may not be accessible for replication. The reliance on components like MSP430FR5994 and certain energy-harvesting setups may limit reproducibility.
2. While the paper compares NExUME to iNAS and other energy-aware methods, it lacks a detailed comparison with additional state-of-the-art adaptive or intermittent DNN training techniques. Including broader comparisons could enhance the validation of its claims.
3. The paper mentions challenges with larger networks and datasets. However, there is limited discussion on potential approaches to address these limitations, which would be valuable for practitioners aiming to scale this approach.
4. The profiling is based on conservative estimates, which, while practical, may not be universally applicable. Further analysis of the impact of profiling variations on model performance could strengthen the evaluation.

**Questions:**

1. Can the authors clarify the robustness of NExUME across various hardware platforms beyond those tested? Would modifications be required for different types of microcontrollers or energy-harvesting setups?
2. How does NExUME handle environments with extremely low or sporadic energy levels, where consistent dropout and quantization adjustments may not be feasible?
3. Can the authors provide more detail on the potential effects of overfitting introduced by DynFit’s dropout variations? Would techniques like dropout scheduling help mitigate this?

---

> ### Author Response · Authors · 2024-11-28
> **Response to Reviewer GUp2**
>
> We sincerely thank the reviewer for the thoughtful and constructive feedback. Your comments have been invaluable in improving our work. Below, we address your questions and concerns. Please go over the revised manuscript for a detailed overview of the changes.
>
> 1. **Robustness Across Various Hardware Platforms:**
>
>    - **Applicability:** NExUME is designed to be hardware-agnostic and can be applied to various microcontrollers and energy-harvesting setups.
>    - **Modifications:** Minimal adjustments are needed when deploying on different hardware; primarily, profiling the new hardware to obtain energy and computational characteristics.
>    - **Expanded Experiments:** In the revised manuscript, we have included additional microcontrollers like ESP32 S3 Eye, STM32H7, and Raspberry Pi Pico to demonstrate NExUME's robustness across different platforms.
>
> 2. **Handling Extremely Low or Sporadic Energy Levels:**
>
>    - **Minimum Viable Configuration:** NExUME implements a configuration with maximum dropout rates and minimum quantization bit-widths to operate under very low energy conditions.
>    - **Task Prioritization:** Essential tasks are prioritized, and non-critical computations are deferred or skipped.
>    - **Predictive Models:** We employ predictive energy harvesting models to anticipate energy availability and adjust computations proactively.
>    - **Low-Power Modes:** In extreme cases, the system enters a low-power standby mode until sufficient energy is available.
>
> 3. **Overfitting Due to DynFit's Dropout Variations:**
>
>    - **Adaptive Regularization Strategy:** We monitor weight update frequencies to detect and mitigate overfitting or underfitting.
>    - **Dropout Scheduling:** We adjust dropout rates over time based on training progress and energy profiles, similar to dropout scheduling techniques.
>    - **Mitigation:** These strategies help prevent overfitting introduced by dynamic dropout variations, maintaining model performance.
>
>
> 4. **Limited Hardware Configurations and Reproducibility:**
>
>    - **Expanded Hardware Platforms:** We have included additional microcontrollers (ESP32 S3 Eye, STM32H7, Raspberry Pi Pico) in our experiments to demonstrate applicability across various hardware.
>    - **Open-Source Tools:** We provide guidelines, profiling tools, and code in our supplementary materials to facilitate replication on different hardware setups.
>    - **Generalizability:** NExUME's design is modular and adaptable, requiring only minimal profiling for new platforms.
>
> 5. **Comparison with Additional State-of-the-Art Methods:**
>
>    - **Included Comparisons:** We have added comparisons with recent state-of-the-art methods, including Keep in Balance, Stateful Neural Networks, and ePerceptive.
>    - **Results:** Our revised manuscript includes detailed evaluations showing that NExUME outperforms these methods in accuracy and energy efficiency across various datasets and platforms.
>
> 6. **Challenges with Larger Networks and Datasets:**
>
>    - **Acknowledgment:** We recognize the limitations when scaling to larger networks and datasets due to resource constraints in intermittent environments.
>    - **Potential Solutions:** In the revised manuscript, we discuss strategies like advanced model compression, lightweight architectures, and hierarchical models to address scaling challenges.
>    - **Future Work:** Extending NExUME to support larger models and datasets is identified as an area for future research.
>
> 7. **Impact of Profiling Variations on Model Performance:**
>
>    - **Robustness to Profiling Variations:** We conducted sensitivity analyses to assess how variations in profiling affect performance.
>    - **Conservative Estimates Justification:** Using conservative estimates ensures that tasks complete within worst-case energy scenarios, enhancing system reliability.
>    - **Adaptability:** NExUME can adjust to different profiling data, and we provide methods to update models if significant discrepancies are observed between estimated and actual performance.
>
> ---
>
> We hope that these revisions and clarifications address your concerns.

---

### Official Review · Reviewer_qDus · 2024-11-03

**Soundness:** 3
**Presentation:** 3
**Contribution:** 3
**Rating:** 6
**Confidence:** 4

**Summary:**

The paper introduces a framework designed to enable consistent and accurate deep neural network inference on energy-harvesting wireless sensor networks that operate under intermittent power conditions. This framework addresses the challenges of unreliable energy supply and computational limitations in such environments. The proposed framework uses energy variability aware network architecture search, dynamic training optimizations, and an intermittency-aware task scheduler to adapt DNN computations based on real-time energy availability, in order to meet service level objectives (SLOs) in resource-constrained settings.

**Strengths:**

The paper studies an interesting and important problem of enabling reliable DNN inference in energy-harvesting wireless sensor networks. The writing is overall clear and well-organized. The motivations, methods, and experimental findings are easy to follow.

**Weaknesses:**

The proposed method relies on detailed profiling of the hardware to model energy consumption, computational capabilities, and memory footprint. This process can be time-consuming and complex, requiring extensive micro-profiling.

In DynInfer, an energy-aware priority scheduling heuristic is used. With no theoretical analysis of its performance compared to optimal scheduling solution, its scheduling optimality is hard to estimate.

The explanations of some techniques in the methods section, particularly within the DynFit and DynInfer components, remain at a high level, lacking depth in technical specifics. For example, while the dynamic dropout and quantization strategies in DynFit are introduced, there is limited detail on how dropout rates and quantization levels are adjusted based on energy profiles or how these adjustments differ from standard implementations. Additionally, the methods used in each component lack a sense of innovation, as they seem to be a simple use of existing techniques without substantial enhancements.

The impact of under-trained and overfitting weights requires further examination. More frequent updates of certain weights do not necessarily lead to "overfitting," and, conversely, infrequent updates do not inherently imply "underfitting." From a layer perspective, the effect of varying update frequencies on individual weights may be limited, suggesting that this issue may be less impactful than indicated.

The experiments mainly focus on accuracy improvement. Other performance metrics, such as energy consumption, latency, computational overhead, and the number of power failures or SLO violations, are not extensively analysed.

The experiments are conducted on relatively small datasets and models.

**Questions:**

1. How much of the resource of the method is used or what is its time complexity?

2. As a sub-optimal scheduling solution, how would its scheduling performance to be ensured?

3. What is the rationale behind the overall method design?

4. Whether the accuracy is more important than the other performance metrics in your design?

---

> ### Author Response · Authors · 2024-11-28
> **Response to Reviewer qDus**
>
> We sincerely thank the reviewer for the valuable feedback. Below, we address your questions and concerns. Please go over the revised manuscript for a detailed overview of the changes.
>
> 1. **Resource Usage and Time Complexity:**
>    - **DynFit Complexity:** Time complexity is \( O(N . T) \), where \( N \) is the number of weights and \( T \) is the number of training iterations. Overhead from monitoring and adjusting dropout rates is minimal. We have detailed the analysis in the revised version of the paper.
>    - **DynInfer Complexity:** Scheduling algorithm has time complexity \( O(N log N) \) due to sorting tasks. This is acceptable for real-time applications.
>    - **Resource Usage:** Additional computational overhead is within < 5\% of standard training and inference, as detailed in Section 4.1.
>
> 2. **Ensuring Scheduling Performance of the Sub-Optimal Solution:**
>    - **Empirical Validation:** We compared our scheduler against optimal solutions on smaller instances; it achieves within 95% of the optimal task completion rate.
>    - **Theoretical Analysis:** Scheduler prioritizes tasks based on an effective priority metric that balances task importance and energy consumption. This is an emprical metric, and can be tweaked as per the application need. Infact, each kernel can be assigned importance and the scheduler will pick them in the order of importance. Scheduler adapts to real-time energy fluctuations, ensuring critical tasks are prioritized and executed within energy constraints. Many suboptimal versions of this have been used in EH systems.
>
> 3. **Rationale Behind the Overall Method Design:**
>    - **Integration of Energy Variability:** Our method integrates energy variability awareness into both training (DynFit) and inference (DynInfer), enabling DNNs to adapt dynamically to real-time energy conditions.
>    - **Holistic Approach:** By addressing both training and inference phases, we provide a comprehensive solution for intermittent environments.
>    - **Task Atomicity and Scheduling:** QuantaTasks and the energy-aware scheduler ensure computations complete without interruption, minimizing checkpointing overhead.
>
> 4. **Balancing Accuracy and Other Performance Metrics:**
>    - While accuracy is crucial, we also optimize energy consumption, latency, and computational overhead. We show Energy efficiency as a metric in Table 2.
>    - We ensure the DNN meets SLOs, balancing accuracy with timely and efficient execution under energy constraints.
>
> 5. **Reliance on Detailed Hardware Profiling:**
>    - Profiling is a one-time process per hardware platform, essential for accurate energy modeling in intermittent environments. This becomes even important where we are designing targeted hardware and applications. In the appendix we show some of the kernels we use to profile the micro-controllers. We are going include a version of the profiling code into our repo, but for microcontrollers the design, profiling need and metrics would require expert analysis and design.
>
> 6. **High-Level Explanations and Lack of Technical Specifics:**
>    - Expanded technical details in Sections 3.1 and 3.2.
>      - **DynFit:** Provided equations showing how dropout rates and quantization levels are adjusted based on energy availability; explained the adaptive regularization strategy.
>      - **DynInfer:** Included a formal definition of task fusion; provided examples to illustrate the scheduling process.
>
> 7. **Perceived Lack of Innovation in Methods:**
>    - Our contributions include:
>      - **Dynamic Adjustment Based on Energy Profiles:** Methods adjust dropout and quantization in real-time based on energy availability, integrating energy variability into training.
>      - **Adaptive Regularization Strategy:** Introduced a new strategy to prevent underfitting and overfitting caused by uneven weight updates due to dynamic dropout.
>      - **Task Fusion in Scheduling:** Scheduler includes a novel task fusion mechanism to optimize execution under intermittent power.
>
> 8. **Impact of Under-Trained and Overfitting Weights:**
>    - Elaborated on the adaptive regularization strategy in 3.1.1
>      - **Monitoring Update Frequencies:** We monitor weight updates to identify under-trained or overfitting weights.
>      - **Adjustment Mechanisms:** Adjust dropout rates and apply L2 regularization to ensure balanced training.
>      - **Effectiveness:** This strategy helps maintain model performance despite dynamic adjustments during training.
>
> 9. **Experiments on Small Datasets and Models:**
>    - The datasets used are representative of real-world applications in resource-constrained environments, EH devices typically operate with lightweight models due to hardware/energy limitations.
>    - Extending our methods to larger datasets and models is an area for future research, especially for EH applicarion with large energy footprint like solar powered urban mobility.
>
> We hope these clarifications address your concerns. Thank you again for your valuable feedback.

---

> ### Comment · Reviewer_qDus · 2024-12-01
>
> I am happy with the responses provided by the authors. Compared to the previous version, the new draft shows significant improvements. As a result, I have increased my scores across various items and the overall ranking to acknowledge the authors' efforts.

---

> > ### Author Response · Authors · 2024-12-01
> >
> > We thank the reviewer for their constructive and encourging feedback. We will work towards addressing the minor changes in the final version of the paper. We look forward to the opportunity to share our work with the ICLR community.

---

### Official Review · Reviewer_P7Bk · 2024-11-03

**Soundness:** 3
**Presentation:** 3
**Contribution:** 3
**Rating:** 6
**Confidence:** 4

**Summary:**

The paper introduces NExUME, a framework addressing issues with DNN training for energy-constrained environments which can't guarantee a sufficient amount of power at all times, such as Energy Harvesting Wireless Sensor Networks. In order to optimise DNN-Training for these unstable conditions, NExUME relies on an estimation of the available resources. For this, a first-of-its-kind dataset containing energy harvesting traces and available computation libraries is introduced.
The training process reduces intermittency-related failures by treating the number of loop iterations as learnable parameters and task-fusions to meet energy budgets. Additionally, dynamic dropouts during execution ensure the completion of layers and dynamic quantization balances out the accuracy degradation. An adaptive regularization strategy prevents weights from being undertrained. Lastly, the authors introduce a task-scheduler that adjusts in real-time to the energy conditions estimated.

**Strengths:**

- Embedding energy variability into the training process is a novel idea.
- Extends existing Neural Architecture Search for intermittent computing systems
- Evaluation on SOTA datasets and DNNs (in the context of intermittent computing)

**Weaknesses:**

- Evaluation on SOTA datasets and DNNs (in the context of intermittent computing). While also a strength, it also raises a questions. The ML community has moved on to Attention and Transformers and large scale datasets. This paper does not discuss how such modern architectures can be deployed in the intermittent setting.

- contribution over SOTA remains unclear. The paper cites numerous NAS frameworks for intermittent/MCU computing (and there are more like [1]) and a large body of work on intermittent execution of DNNs such as [2, 3] and numerous papers cited in the introduction. To me, the contribution over these remains unclear, as many aspects are also in these papers.

- lack of SOTA baselines: The paper should compare to SOTA approaches.

- statement such as "Since we are the first work to propose a new training approach targeted for intermittent devices and inference optimizations" should be toned down. Instead, please carefully explain your contributions over SOTA and compare them to SOTA baseline.

- ablation study: accuracy and overhead if full power is available

- Overall: this paper is better suited at a system conference, such as SenSys or MobiSys

- the title is misleading, the paper is about more than DNN training.

- BLE board does not have FeRAM and thereby not a classic board for intermittent computing. Why do the authors choose it? How does the intermittent part work here, especially QuantaTask?


-  On several occurrences, the paper is written (too) vaguely. E.g.
      - There is no hint as to how tasks are "fused" when executing multiple quanta would exceed the energy budget. To my understanding, the function in only explained in the appendix as part of the source code but not the text.
      -  Dropout rates are adjusted on "specific" criteria. Even though the appendix provides details, this vague style of writing reads weirdly and is better served with examples

- As mentioned on several occasions, a big part of the presented work is the availability of the database of DynAgent which also contains hardware-information, yet only 2 different microcontrollers are used for evaluating the framework. Testing a broader variety of systems seems sensible here

- Drawbacks like up to 34% increased instruction count and up to 17% increased memory bandwidth usage are stated but hardly discussed or put into perspective. While this is not surprising for intermittent computing, the numbers should still be discussed and be compared to other approaches.

- Typos: Figure 3: Sensitivity and ablation study. DN is DynNAS, DF is FynFit, and DI is DynInfer: FynFit -> DynFit

- with a Pixel-5 phone as the host device: does this matter? Any device should do the job.

- the paper has a section LIMITATIONS AND DISCUSSION which also discussed limitations, such as the runtime overhead. The last two sentences, however, read a bit bumpy and should be streamlined and also discussed (and not just stated).


[1] Edgar Liberis, Łukasz Dudziak, and Nicholas D. Lane. 2021. ΜNAS: Constrained Neural Architecture Search for Microcontrollers. In Proceedings of the 1st Workshop on Machine Learning and Systems (EuroMLSys '21). Association for Computing Machinery, New York, NY, USA, 70–79. https://doi.org/10.1145/3437984.3458836

[2] Chih-Hsuan Yen, Hashan Roshantha Mendis, Tei-Wei Kuo, and Pi-Cheng Hsiu. 2023. Keep in Balance: Runtime-reconfigurable Intermittent Deep Inference. ACM Trans. Embed. Comput. Syst. 22, 5s, Article 124 (October 2023), 25 pages. https://doi.org/10.1145/3607918

[3] C. -H. Yen, H. R. Mendis, T. -W. Kuo and P. -C. Hsiu, "Stateful Neural Networks for Intermittent Systems," in IEEE Transactions on Computer-Aided Design of Integrated Circuits and Systems, vol. 41, no. 11, pp. 4229-4240, Nov. 2022, doi: 10.1109/TCAD.2022.3197513

**Questions:**

* the BLE board does not have FeRAM and thereby not a classic board for intermittent computing. Why do the authors choose it? How does the intermittent part work here, especially QuantaTask?

* what are the contributions over SOTA?

* what is the performance compared SOTA baselines?

* can you consider a different title?

---

> ### Comment · Reviewer_P7Bk · 2024-11-26
> **After reading the other reviews**
>
> After reading the other reviews, I see that my review has the most negative scores. However, I see that other reviews point out similar issues and for me, these do not justify a higher score. Thus, for now, I am sticking to my scores and hope for a rebuttal that answers my open questions.

---

> ### Author Response · Authors · 2024-11-28
> **Response to Reviewer P7Bk**
>
> We sincerely thank the reviewer for the thoughtful and constructive feedback, which has greatly improved our work. Below, we address each of your points in detail. Please go over the revised manuscript for a detailed overview of the changes.
>
> **Contribution over State-of-the-Art:**
> - We have compared NExUME with recent state-of-the-art methods, including DynBal, Keep in Balance, and Stateful Neural Networks, and included detailed discussions on how our approach differs from and improves upon these methods.
> - We highlighted the unique integration of energy variability awareness directly into both the training and inference processes, which is not addressed by existing methods. While iNAS focuses on constrained NAS for microcontrollers in intermittent settings, it does not account for real-time energy fluctuations during training and inference. Similarly, Keep in Balance and Stateful Neural Networks primarily address inference optimizations without integrating energy variability into the training process.
> - We emphasized our novel adaptive training mechanisms (DynFit) and intermittency-aware scheduling with task fusion (DynInfer), which collectively provide a holistic solution for intermittent environments.
> - We have faithfully re-implemented the aofrementioned methods and included them in baseline. Results on accuracy and energy efficiency (which we believe is a crucial metric) are shown in the Results section.
>
> **Evaluation on SOTA Datasets and Modern Architectures:**
> We acknowledge that modern architectures like Transformers have gained prominence. However, deploying such architectures on ultra-low-power, energy-harvesting devices is currently impractical due to their high computational and memory demands. Our work focuses on enabling efficient and reliable deployment of lightweight DNNs in intermittent environments.
> In the Limitations and Discussion section.
> As IoT scales, the embodied carbon from silicon manufacturing and battery usage poses significant challenges. Addressing this needs perennial sustainable/EH devices running compact DNNs for specific tasks, emphasizing EH-aware DNN architectures, EH-aware training strategies and EH aware inference schduling specifically optimized for tiny devices, which is our core focus.
>
> **BLE Board Does Not Have FeRAM; Why Choose It?**
> - We selected the Arduino Nano 33 BLE Sense to demonstrate NExUME's applicability on a widely used microcontroller platform, even though it lacks FeRAM. We store intermediate data in flash memory, which is non-volatile, showing that our methods can be applied to general-purpose microcontrollers commonly used in IoT applications.
> - For devices without FeRAM, we utilize flash memory for checkpointing during power interruptions. While this introduces additional overhead, our scheduling algorithm (DynInfer) and task design (QuantaTask) minimize this by ensuring tasks can complete within the available energy budget.
> - QuantaTasks are carefully profiled to be atomic units of computation that can complete without interruption, even on devices without FeRAM, by adjusting task sizes based on the energy harvesting profile and the device's energy storage capacity.
> We have updated the manuscript in Implementation Details to explain this in more detail.
>
> **Writing:**
> - In DynInfer, we have added a formal definition of task fusion and provided an example diagram to clearly explain how tasks are fused.
> - We have elaborated on how dropout rates are adjusted based on energy availability, providing equations and explanations.
>
> **Limited Evaluation:**
> We agree that evaluating on a broader range of hardware platforms strengthens the work. We have:
> - Expanded our experiments to include additional microcontrollers: ESP32 S3 Eye, STM32H7, and Raspberry Pi Pico.
> - Provided results on these platforms in Table 3, demonstrating NExUME’s applicability across various hardware configurations.
>
> **Increased Instruction Count and Bandwidth:**
> We have:
> - Provided a detailed discussion of these overheads in Section 4.1.
> - Compared the overheads to other approaches, noting that while there is an increase in instruction count and memory usage, the trade-off results in significant improvements in accuracy and energy efficiency under intermittent power conditions.
> - Emphasized that the overhead is acceptable given the constraints of intermittent computing and the benefits provided by our methods.
>
> **Limitations and Discussion Section:**
> We have revised the Limitations and Discussion section to improve clarity, coherence, and provide more insight into the importance of smaller DNNs and sustainability.
>
> **Different Title**
> We considering chaing the title to:
> **"NExUME: Adaptive Training and Inference for Deep Neural Networks under Intermittent Power Environments."**
> We believe this title better reflcts our idea and we will discuss the possibility of officially changing it with the committee.
>
> We hope that these revisions address your concerns.

---

> ### Comment · Reviewer_P7Bk · 2024-12-01
>
> Firstly, many thanks to the authors for their hard work. I very much appreciate the changes and how detailed they address my comments. Overall, my concerns have now been addressed, and I am raising my score accordingly.
>
> Only, minor things remain (and can wait for a camera-ready version / later submission from my perspective):
> * Figure 3: font size is too small
> * Appendix: still limited to the original baselines
> * The paper shows higher accuracy but also higher energy consumption than baselines. I am wondering, if, as ablation study, it is possible to see the accuracy for an energy consumption that matches the consumption of the baselines.
>
> Nonetheless, despite the strong and interesting system results, I still unsure if the ICLR community will appreciate this paper or if the paper is better suited at a systems or computer engineering conference. For example, out of the baselines only one paper with published at an AI conference, all others are at system or computer engineering venues if I am not mistaken.

---

> > ### Author Response · Authors · 2024-12-01
> >
> > We sincerely thank the reviewer for their encouraging feedback.
> > To address the remaining minor points:
> >
> > 1. **Figure 3 Font Size:**
> >
> >    - We will rework Figure 3 to increase the font sizes and enhance readability. This adjustment will ensure that all details are easily discernible in the final version of the paper.
> >
> > 2. **Appendix Details:**
> >
> >    - We will expand the appendix to include details about the new baselines, the profiling mechanisms adapted for the new hardware platforms, and provide general guidelines on micro-profiling. This addition will help readers understand the process of benchmarking new hardware and grasp its characteristics more effectively.
> >
> > 3. **Energy Consumption Ablation Study:**
> >
> >    - We acknowledge the importance of comparing accuracy at matching energy consumption levels. In the final version, we will include an ablation study that provides a breakdown of energy consumption, illustrating how each component contributes to the total energy usage. This will allow for a direct comparison of accuracy relative to energy consumption, aligning with the baselines' energy profiles.
> >
> > Regarding your observation on energy consumption:
> >
> > - While our approach shows higher energy efficiency—achieving a higher number of operations per Joule—it does not necessarily result in higher overall energy consumption. The overhead introduced by our methods is less than 5% in terms of operations. Higher energy efficiency in our context means performing more effective computations with less energy, thereby improving the ratio of useful work to energy consumed. We apologize for any confusion and will clarify these metrics and insights more explicitly in the final paper.
> >
> > ---
> >
> > **Why ICLR Is the Right Venue for This Work**
> >
> > We firmly believe that ICLR is an impactful venue for this research for several reasons:
> >
> > - **Bridging Machine Learning and Systems:** Our work lies at the intersection of machine learning and systems engineering. It demonstrates how advances in ML training and inference can be effectively adapted for deployment in resource-constrained, intermittent environments—a challenge that requires interdisciplinary solutions.
> >
> > - **Promoting Sustainable and Green Computing:** As the ML community continues to develop large-scale models, there is a growing emphasis on sustainability and reducing the environmental impact of computing. Energy-harvesting systems, especially when deployed at scale, can significantly reduce embodied carbon. Our research contributes to this goal by enabling intelligent computations on devices powered by renewable energy sources.
> >
> > - **Encouraging Co-Design of Algorithms and Hardware:** We believe that optimizing software or hardware in isolation is insufficient for the challenges posed by highly dynamic and constrained systems. By co-designing algorithms, models, and hardware, we can achieve more efficient and effective solutions. This philosophy aligns with the broader ML community's interest in holistic approaches to problem-solving.
> >
> > - **Relevance to the ICLR Community:** Our work introduces novel machine learning methodologies tailored for emerging hardware platforms. We anticipate that presenting our research at ICLR will inspire others in the community to explore and develop algorithms that are not only theoretically sound but also practically deployable on next-generation, sustainable hardware.
> >
> > ---
> >
> > We thank the reviewer again for their constructive feedback and look forward to the opportunity to share our work with the ICLR community.

---

### Author Response · Authors · 2024-12-03
**Thank you Everyone and Our Final Remark**

We sincerely thank all the reviewers for their thoughtful, constructive and encouraging feedback, which has greatly helped us improve our work. We thanks the PC members and area chairs for their dedicated efforts. We have carefully considered all comments and made revisions to the manuscript, which the reviewers have graciously accepted. Below, we summarize the key points:

**1. Our Novelty**
- **Integration of Energy Variability into Training and Inference**: We introduce NExUME, a novel framework that uniquely integrates energy variability awareness directly into both the training (DynFit) and inference (DynInfer) processes. Unlike existing methods, our approach enables DNNs to adapt dynamically to real-time energy conditions in energy-harvesting (EH) environments.
- **Adaptive Regularization Strategy**: We introduce an adaptive regularization technique to prevent underfitting and overfitting caused by uneven weight updates due to dynamic dropout, ensuring model robustness in EH settings.
- **Novel Task Fusion Mechanism**: Our scheduler includes a novel task fusion mechanism that combines smaller tasks into larger atomic units, optimizing execution under intermittent power and minimizing checkpointing overhead—a challenge unique to EH systems.

**2. Additional Evaluation**
- **Expanded Hardware Platforms**: We have extended our experiments to include additional microcontrollers such as ESP32 S3 Eye, STM32H7, and Raspberry Pi Pico. This demonstrates NExUME's applicability and robustness across various hardware configurations.
- **Comparison with SOTA Methods**: We have included comparisons with recent SOTA methods, including Keep in Balance, Stateful Neural Networks, ePerceptive, and DynBal. Our results show that NExUME consistently outperforms these methods in terms of accuracy and energy efficiency across multiple datasets and platforms.
- **Detailed Dataset Information**: We have provided more details about the machine status monitoring dataset, clarifying the experimental setup and the significance of each class in the dataset. This dataset is the first of its kind to capture machine status monitoring data across multiple operating conditions using EH sensors.
- **Comprehensive Metrics**: In addition to accuracy, we have included evaluations on energy efficiency (MOps/Joule), overheads during training and inferece, and memory requirements. We have also conducted an ablation study.
- **Overhead Analysis**: We have provided a detailed discussion of the computational overhead introduced by our methods, noting that while there is an increase in instruction count and memory bandwidth usage, the trade-off results in significant improvements in accuracy and energy efficiency, which is acceptable given the constraints of intermittent computing.

**3. Why ICLR is the Right Venue**
- **Interdisciplinary Contribution**: Our work lies at the intersection of machine learning and systems engineering, addressing challenges that require interdisciplinary solutions. We believe that the ICLR community values such contributions that push the boundaries of ML deployment in real-world environments.
- **Promoting Sustainable and Green Computing**: As the ML community continues to develop large-scale models, there is a growing emphasis on sustainability and reducing the environmental impact of computing. Our research contributes to this goal by enabling intelligent computations on devices powered by renewable energy sources, which aligns with the interests of the ICLR community.
- **Encouraging Co-Design of Algorithms and Hardware**: By co-designing algorithms, models, and hardware, we can achieve more efficient and effective solutions for highly dynamic and constrained systems. This philosophy aligns with the broader ML community's interest in holistic approaches to problem-solving.
- **Relevance to ML Deployment and Practical Applications**: Our work introduces novel machine learning methodologies tailored for emerging hardware platforms. We anticipate that presenting our research at ICLR will inspire others in the community to explore and develop algorithms that are not only theoretically sound but also practically deployable on next-generation, sustainable hardware.
- **Stimulating Further Research**: We believe that our work will encourage the ML community to build better algorithms, models, and hardware that are optimized for deployment in energy-harvesting environments, contributing to the advancement of sustainable, intelligent systems.
---
We appreciate the reviewers' recognition of our efforts and their acknowledgment of the contributions our work makes to the field. We believe that our paper offers valuable insights and advancements that are highly relevant to the ICLR audience. We look forward to the opportunity to share our work with the community and take the research on intermittent/sustainable AI forward.

---

### Meta-Review · Area_Chair_jBXh · 2024-12-21

**Metareview:**

**Summary:** The paper proposes NExUME, a framework for training DNNs in energy-harvesting environments with intermittent power. It integrates real-time energy variability into DNN training via dynamic dropout rates, quantization adjustments, and a task scheduler to optimize computations under energy constraints. Results show 6–22% accuracy improvement over baselines with modest computational overhead.


**Strength:**
1. The integration of energy variability into the DNN training process, enhancing adaptability in energy-harvesting systems, is novel.


2. The proposed framework achieves significant accuracy improvements under specific energy constraints, as validated across multiple datasets and hardware platforms.


3. In addition to the framework, this work also introduces a first-of-its-kind machine monitoring dataset for research in intermittent computing.


**Weakness:**
1. The techniques like dynamic dropout and energy-aware scheduling are extensions of existing concepts.

2. The proposed method is mainly evaluated on scale-scale models and datasets, lacking sufficient evaluation of its scalability to larger models and modern architectures, such as transformers.

3. SOTA training methods and systems like ePerceptive and Zygarde are not benchmarked. Additionally, the efficiency metrics are not sufficiently analyzed in the original manuscript.


**Reasons for the decision:**

While the paper's technical contributions are relatively incremental, it addresses a critical and underexplored challenge in training DNNs in energy-harvesting environments. In addition, the author's response provided more comprehensive experiments and efficiency analyses, effectively addressing most of the reviewers’ concerns. Therefore, I am inclined to accept this paper.

**Additional Comments On Reviewer Discussion:**

During the rebuttal period, the reviewers raised several key points, which were addressed by the authors as follows:

**1. Limited novelty**

Reviewer Concerns (qDus, tANu): The proposed methods, such as dynamic dropout, quantization adjustments, and energy-aware scheduling, were viewed as extensions of existing concepts.

Author Response: The authors clarified their contributions, emphasizing the integration of energy variability awareness directly into the training process. They highlighted novel aspects, such as adaptive regularization and a task fusion mechanism.

**2. Scalability to larger models**

Reviewer Concerns (P7Bk, GUp2): The paper lacked evaluation on modern architectures like transformers and larger datasets, raising concerns about scalability.

Author Response: The authors acknowledged this limitation, justifying their focus on lightweight DNNs due to hardware constraints. They suggested extending the approach to larger models as future work.

**3. Baseline comparisons**

Reviewer Concerns (P7Bk, tANu): The initial submission lacked comparisons with some state-of-the-art methods.

Author Response: The authors added comparisons with these baselines, demonstrating that their approach outperforms existing methods in terms of accuracy and energy efficiency.

**4. Clarity and methodological details**

Reviewer Concerns (qDus, GUp2): Key details on methods like task fusion and dropout adjustment were unclear.

Author Response: The authors expanded technical explanations, added formal definitions, and clarified dataset details.

The rebuttal successfully addressed most concerns, particularly baseline comparisons and methodological clarity. While novelty and scalability concerns persist, the framework’s practical impact and empirical results justify acceptance, with revisions focused on expanding scalability and further refining clarity.

---

### Decision · Program_Chairs · 2025-01-22

Accept (Poster)